# Morphogenetic forces planar polarize LGN/Pins in the embryonic head during *Drosophila* gastrulation

**Jaclyn Camuglia[1], Soline Chanet[2], Adam C Martin[1]\***

[1]Biology Department, Massachusetts Institute of Technology, Cambridge, MA, United States; [2]Center for Interdisciplinary Research in Biology (CIRB), Collège de France, CNRS, INSERM, Université PSL, Paris, France

**Abstract** Spindle orientation is often achieved by a complex of Partner of Inscuteable (Pins)/LGN, Mushroom Body Defect (Mud)/Nuclear Mitotic Apparatus (NuMa), Gαi, and Dynein, which interacts with astral microtubules to rotate the spindle. Cortical Pins/LGN recruitment serves as a critical step in this process. Here, we identify Pins-mediated planar cell polarized divisions in several of the mitotic domains of the early *Drosophila* embryo. We found that neither planar cell polarity pathways nor planar polarized myosin localization determined division orientation; instead, our findings strongly suggest that Pins planar polarity and force generated from mesoderm invagination are important. Disrupting Pins polarity via overexpression of a myristoylated version of Pins caused randomized division angles. We found that disrupting forces through chemical inhibitors, depletion of an adherens junction protein, or blocking mesoderm invagination disrupted Pins planar polarity and spindle orientation. Furthermore, directional ablations that separated mesoderm from mitotic domains disrupted spindle orientation, suggesting that forces transmitted from mesoderm to mitotic domains can polarize Pins and orient division during gastrulation. To our knowledge, this is the first in vivo example where mechanical force has been shown to polarize Pins to mediate division orientation.

**\*For correspondence:**
acmartin@mit.edu

**Competing interest:** The authors declare that no competing interests exist.

## Editor's evaluation

This study provides compelling in vivo evidence that mechanical tension emanating from morphogenetic forces during gastrulation orients the spindle at distant sites. The finding that gastrulation-induced forces are required for asymmetric localization of an important and evolutionarily conserved spindle orientation factor, Pins, will be of broad interest to cell and developmental biologists.

## Introduction

Epithelial morphogenesis and homeostasis are influenced by cell division orientation, which can occur along specific axes relative to both the epithelial plane and the polarity of an organ/organism (*Baena-López et al., 2005*; *Kulukian and Fuchs, 2013*; *Lancaster and Knoblich, 2012*; *Morin and Bellaïche, 2011*). Epithelia are tissues composed of cells that are organized in a sheet and held together by junctions, which maintain the integrity of the tissue as well as the apical-basal polarity of cells (*Choi et al., 2013*; *Classen et al., 2005*; *Cox et al., 1996*; *Gumbiner, 2005*; *Harris and Peifer, 2005*; *Tepass, 1997*; *Wang and Margolis, 2007*). Asymmetric cell divisions, in which cellular components, such as existing adherens junctions, are distributed unequally to daughter cells can result in one daughter cell leaving the epithelium. In contrast, symmetric divisions in epithelia divide cellular components equally, and usually results in both daughter cells remaining in the tissue. Orientation of the mitotic

spindle is what determines whether a division is symmetric or not, as spindle orientation determines division orientation (*Cabernard and Doe, 2009*; *Gönczy and Rose, 2005*; *Lechler and Fuchs, 2005*; *Mao et al., 2011*; *Nakajima et al., 2013*). In addition, the spindle orientation with respect to organ/organism axis can promote directional growth and/or stress dissipation in tissues (*Campinho et al., 2013*; *Legoff et al., 2013*; *Mao et al., 2011*; *Mao et al., 2013*; *Wyatt et al., 2015*).

Spindle orientation is often achieved by a complex of Partner of Inscuteable (Pins)/LGN, Mushroom Body Defect (Mud)/Nuclear Mitotic Apparatus (NuMa), Gαi, and Dynein. In this complex, Pins/LGN acts as a linker between Gαi and Mud, which is also bound to a Dynein-Dynactin complex. Dynein's motor activity provides the force to rotate the spindle (*Bergstralh et al., 2013*; *Busson et al., 1998*; *Fielmich et al., 2018*; *Grill and Hyman, 2005*; *Johnston et al., 2009*; *Nguyen-Ngoc et al., 2007*; *Siller and Doe, 2008*). In *Drosophila* neuroblasts, asymmetric division and spindle orientation is dependent on Pins/LGN recruitment to the apical cortex. Apical proteins recruit Inscuteable (Insc), which in turn recruits Pins/LGN and the spindle rotation complex. Pins null mutants exhibit defects in spindle orientation in neuroblasts (*Cabernard and Doe, 2009*; *Schaefer et al., 2000*; *Siller et al., 2006*; *Yu et al., 2006*). In symmetric epithelial cell divisions, spindle orientation also depends on Pins, which is recruited to the lateral cortex by the lateral domain protein, Discs large (Dlg). In other cell types, cell intrinsic signals can localize Pins/LGN (*Kiyomitsu and Cheeseman, 2012*; *Zheng et al., 2013*). Loss of Pins, Dlg, or disrupted recruitment of Pins by Dlg causes aberrant asymmetric cell divisions in epithelia and cell loss (*Bell et al., 2015*; *Bergstralh et al., 2013*; *Carvalho et al., 2015*). Furthermore, in planar cell polarized divisions of the sensory organ precursor (SOP) cells in the *Drosophila* dorsal thorax, asymmetric determinant segregation and spindle orientation depend on planar polarized Pins and E-cadherin localization, respectively (*Bellaïche et al., 2001*; *Le Borgne et al., 2002*).

In addition to molecular cues, mechanical cues also orient cell division. For example, in isolated cell cultures, cell division orientation has been shown to directly respond to stress by aligning with applied forces (*Fink et al., 2011*; *Matsumura et al., 2016*; *Théry et al., 2007*; *Théry et al., 2005*). Cells also respond to force through its influence on cell shape. The connection between cell shape and division axis is known as the long axis rule or Hertwig's rule and was first observed over 100 years ago (*Hertwig, 1884*). However, the mechanisms through which force, cell shape, and molecular cues are interconnected to influence cell division remain an active area of research. In epithelial tissues, cell division/spindle orientation occurs along the tension axis, which can dissipate tension and influence cell packing (*Campinho et al., 2013*; *Gibson et al., 2011*; *Godard et al., 2020*; *Mao et al., 2011*; *Scarpa et al., 2018*; *Wyatt et al., 2015*). In the *Drosophila* pupal notum and *Xenopus,* cell shape has been shown to affect the spindle angle through the influence of molecular cues localized at tricellular junctions, which provide a 'memory' for interphase cell shape (*Bosveld et al., 2016*; *Nestor-Bergmann et al., 2019*). In contrast, in the *Drosophila* oocyte, follicular epithelium, and parasegmental boundaries and cultured MDCK cells, molecular cues, like Pins/LGN and Mud/NUMA, do not localize to tricellular junctions, implying that cells use additional strategies to promote force-oriented spindle rotation, in some contexts (*Finegan et al., 2019*; *Gloerich et al., 2017*; *Scarpa et al., 2018*; *Yu et al., 2006*). For example, mechanical cues have been shown to recruit the Pins/LGN machinery to adherens junctions in MDCK cells (*Hart et al., 2017*). Mitotic rounding forces have also been shown to be important for proper spindle orientation and morphology (*Chanet et al., 2017*; *Kunda et al., 2012*). The combinatorial action of mechanical and molecular cues used to orient spindles is incompletely understood. Here, we use the *Drosophila* embryo as a powerful model organism that is uniquely tractable – exhibiting mirror-symmetric domains that undergo highly stereotyped oriented divisions – to uncover the relationship between mechanical and molecular cues that coordinate division orientation and morphogenetic movements in an embryo.

During development, the early *Drosophila* embryo undergoes 13 cycles of synchronous nuclear divisions with no intervening gap phases. After these 13 cycles of nuclear division, embryonic cells arrest in G2. Cycle 14 divisions occur in a highly stereotyped pattern defined by *string* (*Drosophila* Cdc25 homologue) expression in pockets of cells across the embryo. These pockets of cells that divide together are termed mitotic domains (*Edgar and O'Farrell, 1989*; *Foe, 1989*). Some of these domains have been observed to have oriented divisions, but the mechanism underlying division orientation is unknown (*da Silva and Vincent, 2007*; *Hartenstein and Campos-Ortega, 1985*; *Hartenstein et al., 1987*; *Stern et al., 2022*; *Wang et al., 2017*). We found that spindle orientation is a dynamic process in which spindles are not assembled in position, but rather rotate to achieve orientation in

mitotic domains 1, 3, and 5 (MDs 1, 3, and 5) in the dorsal 'head' and are oriented in the direction of eventual tissue movement/elongation. We discovered that these oriented symmetric divisions and those of MD14, which had previously been shown to exhibit force-dependent oriented symmetric divisions (*Wang et al., 2017*), require planar polarized Pins.

## Results

### Early mitotic domains in the *Drosophila* head exhibit spindle rotation and planar polarized division

Located on the dorsal side of the *Drosophila* embryo 'head', MDs 1, 3, and 5 undergo division during gastrulation, 200 min after egg deposition (AED). MD3 is the anterior most mitotic domain and comprises around 30 cells, MD1 is the next most anterior domain and consists of two mirrored clusters of around 40 cells, and MD5 is located just anterior to the cephalic furrow and consists of two mirrored clusters containing around 30 cells (*Figure 1A and B*).

We initially visualized division axis in embryos expressing fluorescently tagged non-muscle myosin II (Myo II) to mark cytokinetic rings and fluorescently tagged myristoylation sequence of Gap43 to mark cell membranes. Cytokinetic rings form perpendicular to the division axis. In MDs 1, 3, and 5, cytokinetic rings are oriented roughly perpendicular to the anterior-posterior (AP) axis (*Figure 1C*). To further visualize the division process and assess dynamics during division, we used embryos expressing fluorescently tagged CLIP170 to mark the mitotic spindle (*Figure 1D*, *Figure 1—video 1*). For this paper, we will display all head images with the anterior pole toward the top. To quantify the alignment of the divisions with the AP axis, we measured the angle between the spindle and the AP axis, where an angle of 0 indicates perfect alignment with the AP axis and an angle of 90 indicates alignment with the dorsal-ventral (DV) axis (*Figure 1E*). Spindles were measured after anaphase, when spindle position becomes fixed. MDs 1, 3, and 5 divisions occur almost entirely within 30° of the AP axis (*Figure 1F–H*). Markedly in MD1, the majority of spindles were slightly tilted relative to the AP axis and mirror symmetric, such that spindles on the left side divided within 0 to 30° of the AP axis (*Figure 1C and I*) and spindles on the right side divided within 0 to –30° of the AP axis (*Figure 1D and J*).

Notably, we found that mitotic spindles are not initially oriented with the AP axis and form with no preferential orientation. Spindles then rotated up to 90° to achieve a final orientation (*Figure 1D*). Despite dividing with a mirror-symmetric tilt away from the AP axis, we observed no chirality in spindle rotation; on the left side of MD1 53% of measured spindles rotated counterclockwise and 47% rotated clockwise, on the right side 46% rotated counterclockwise and 54% clockwise. Together these data show that MDs 1, 3, and 5 exhibited oriented divisions and this orientation is achieved by spindle rotation toward an angle ~30° from the AP axis. We next asked what cue(s) could coordinate the spindle rotation and division orientation.

### Planar polarized divisions require polarized Pins localization

Because spindles exhibited a pronounced rotation after assembly, we examined components of the spindle rotation machinery, Pins and Mud. We discovered that MDs 1, 3, and 5 all exhibit planar cell polarity of Pins. The Pins planar cell polarity is dependent on cell cycle state: mitotic cells exhibit planar polarized Pins, whereas Pins is uniformly distributed along lateral membranes of interphase cells (*Figure 2A*, *Figure 2—video 1*). We quantified Pins polarization by modifying previously described polarization quantifications of mitotic cells (*Besson et al., 2015*). We used embryos expressing fluorescently labeled Pins and fluorescent membrane marker Gap43::mCherry, and measured the intensity of Pins::YFP and Gap43::mCherry around the membrane. We found that in MD1, the Pins intensity is highest on the posterior side of the cells and the crescent of Pins intensity peaks just past the posterior-most point of the cell cortex (*Figure 2C and C'*). The ratio of the point of greatest Pins intensity to the average of the two adjacent troughs averaged over three embryos was significantly greater than the same ratio calculated from the Gap43::mCherry signal (peak-to-trough ratio Pins, 1.39±0.1017 and Gap43::mCherry, 1.04±0.0406, mean and SD, p=1.7057e-08), demonstrating that this polarity was not caused by membrane inhomogeneity. Gap43::mCherry also had a more uniform labeling along the membrane in MD cells (*Figure 2B, D and D'*). Though our imaging resolution is not sufficient to determine whether Pins is localized solely on the posterior side of MD1 cells or both

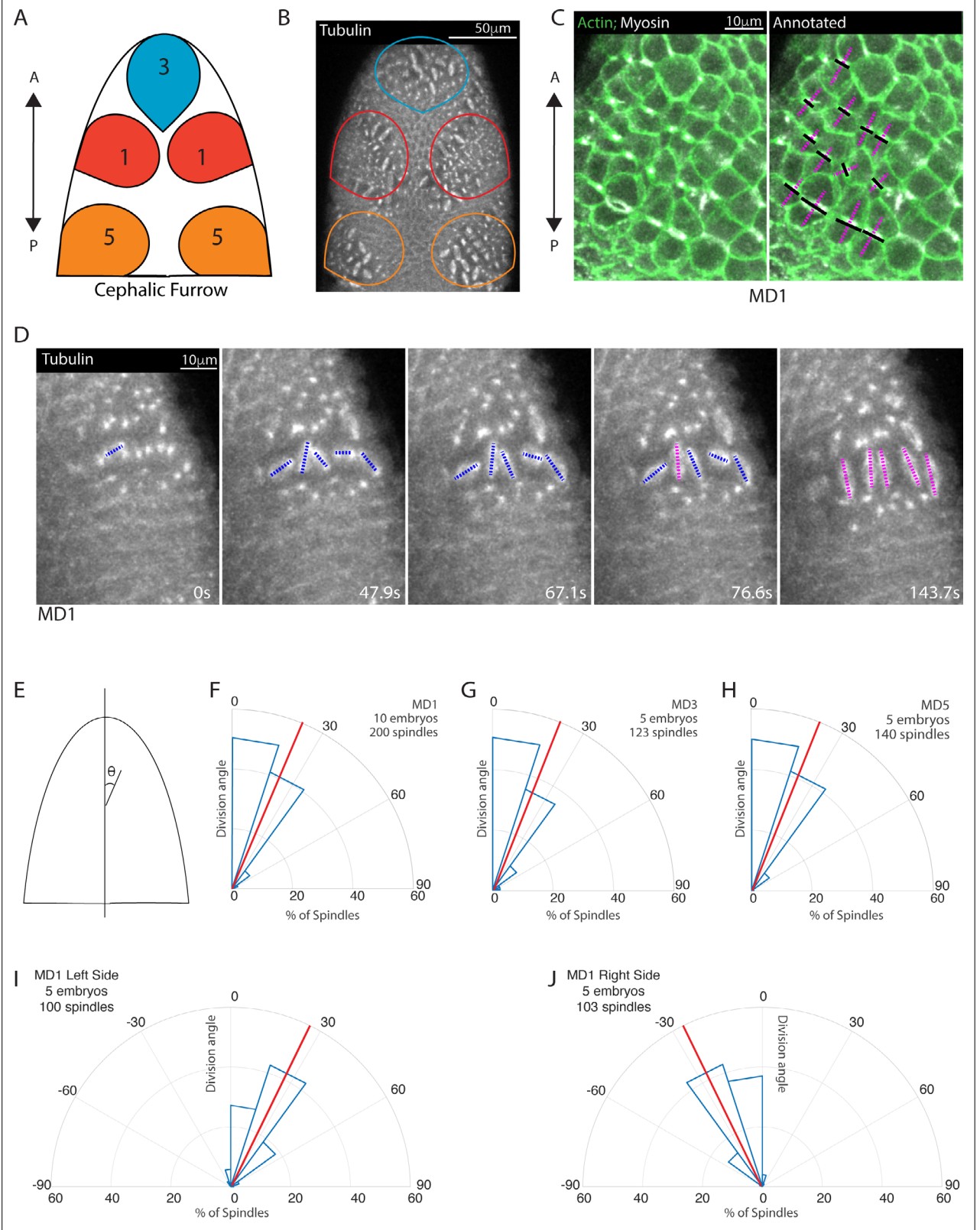

**Figure 1.** Early mitotic domains in the *Drosophila* head exhibit spindle rotation and oriented division. (**A**) Schematic of the dorsal head of a *Drosophila* embryo with mitotic domain 1 (MD1) in red, mitotic domain 3 (MD3) in blue, and mitotic domain 5 (MD5) in orange. Anterior-posterior (AP) axis is indicated by arrow, the top of images will consistently be anterior, as in the schematic. (**B**) Image of the dorsal head of a live *Drosophila* embryo expressing CLIP170::GFP to label microtubules/spindle (grayscale). CLIP170::GFP signal is detected in image. MD1 is outlined in red, MD3 in blue,

*Figure 1 continued*

and MD5 in orange, scale bar is 50 µm. (**C**) MD1 cells exhibit planar polarized cell division. Left, image of MD1 in live embryo showing labeled myosin (grayscale, Sqh::mCherry), to show cytokinetic rings, and labeled F-actin (green, UtrABD::GFP) to show cell cortex, scale bar is 10 µm. Right, same image of MD1 annotated with black lines marking the cytokinetic ring of each dividing cell and magenta dashed lines marking division angle. (**D**) MD1 cells exhibit spindle rotation and alignment. Images are time series of MD1 with CLIP170::GFP marking the mitotic spindle (grayscale), blue lines indicate spindles still rotating and magenta lines represent spindle that have reached their final orientation, scale bar is 10 µm. (**E**) Schematic of division angle analysis. Division angle is taken with respect to the AP axis such that angle of 0 indicates alignment with the AP axis. Note only spindles during or after anaphase, when division angle is fixed, are measured. (**F–H**) Rose plots indicating division angles of MD1 (**F**), MD3 (**G**), and MD5 (**H**). Red lines indicate mean division angle. Mean division angle for MD1 = 21.6°, MD3=20.4°, and MD5=19.0°. For MD1, n=10 embryos and 200 spindles, MD3, n=5, embryos and 123 spindles, and MD5, n=5 embryos and 140 spindles. (**I–J**) Rose plots indicating division angles of left (**I**) or right (**J**) side of MD1. For left side, n=5 embryos and 100 spindles and right side, n=5 embryos and 103 spindles.

The online version of this article includes the following video for figure 1:

**Figure 1—video 1.** Max intensity projection (with Gaussian blur) of control embryo expressing Jup::GFP (to label mitotic spindle) during divisions of mitotic domains (MDs) 1, 3, and 5.

https://elifesciences.org/articles/78779/figures#fig1video1

the posterior and anterior sides when groups of cells round up together, we find that cells on the boundary of MD1 display Pins crescents enriched on the posterior side. Interestingly, from observation of boundary cells, we find Pins was enriched on the posterior side of MDs 1 and 3, but the anterior side of MD5 (*Figure 2—figure supplement 1*).

To ensure Pins polarization was not a consequence of expressing non-endogenous levels of Pins, we fixed and stained embryos to localize endogenous Pins. We found that similar to Pins::YFP, endogenous Pins was planar polarized in mitotic MD1 cells (*Figure 2E–F*). We also looked at components of the spindle rotation machinery complex downstream of Pins, namely Mud. It appears that Mud follows a similar localization pattern as Pins and is polarized in crescents in mitotic cells of MD1, although Mud also strongly labels spindle poles (*Figure 2G*). Based on our live and fixed imaging, we conclude that MDs 1, 3, and 5 cells exhibit planar polarized Pins localization during mitosis.

In addition to MDs 1, 3, and 5, we examined MD14, which has previously been shown to exhibit force-dependent planar polarized divisions (*Wang et al., 2017*). Mitotic cells within MD14 exhibited planar polarized Pins, similar to what we saw for MD1 (*Figure 2H*). Posterior crescents of Pins form in the mitotic cells of MD14, though the crescent is broader than in MDs 1, 3, and 5. The peak-to-trough ratio for Pins in MD14 was significantly different from control Gap43::mCherry (1.22±0.0914 and 1.04±0.1115, respectively, mean and SD, p=3.1726e-06) (*Figure 2H–K'*). Thus, MD14 cells also exhibit planar polarized Pins localization.

We next sought to establish whether there was a connection between Pins polarity and spindle orientation. Using embryos expressing fluorescently tagged tubulin and fluorescently tagged Pins, we monitored the timing of Pins polarization and spindle rotation. We found that Pins polarization preceded spindle rotation and is abolished shortly after anaphase, when the spindle position is finalized (*Figure 3A–B*, *Figure 3—video 1*). The correlated timing between Pins polarization and spindle rotation suggests that Pins is responsible for the spindle achieving its polarized orientation. To determine the relationship between Pins polarity and spindle rotation we compared the angle of maximum Pins intensity to the division angle. We found that there was a strong correlation between angular position of maximum Pins intensity and the anaphase spindle angle in MD1 (*Figure 3C*). Previous studies have shown division orientation often follows the long axis of the interphase cell shape (*Hertwig, 1884*; *Minc et al., 2011*; *Tsou et al., 2003*). To determine whether division angle in the MD1 could also be related to cell shape, we compared the interphase cell long axis angle to the division angle. In contrast to Pins polarity, there was no correlation between interphase cell shape and division angle, indicating the divisions of MD1 do not follow the long axis rule (*Figure 3—figure supplement 1*). Together these data show that division orientation correlates with Pins localization, but not interphase cell shape. We also analyzed Pins and tubulin within MD14, which is located at the ventral midline (*Figure 3D*). Similar to MD1, Pins polarization preceded spindle rotation (*Figure 3E–F*).

To test the functional requirement of Pins polarity, we analyzed spindle orientation in embryos overexpressing a myristoylated Pins. Myristoylation covalently attaches a myristoyl group to a protein so the protein associates with the plasma membrane in an unregulated manner. Overexpression of myristoylated Pins, but not wild-type Pins, caused Pins to be uniformly distributed along the membrane

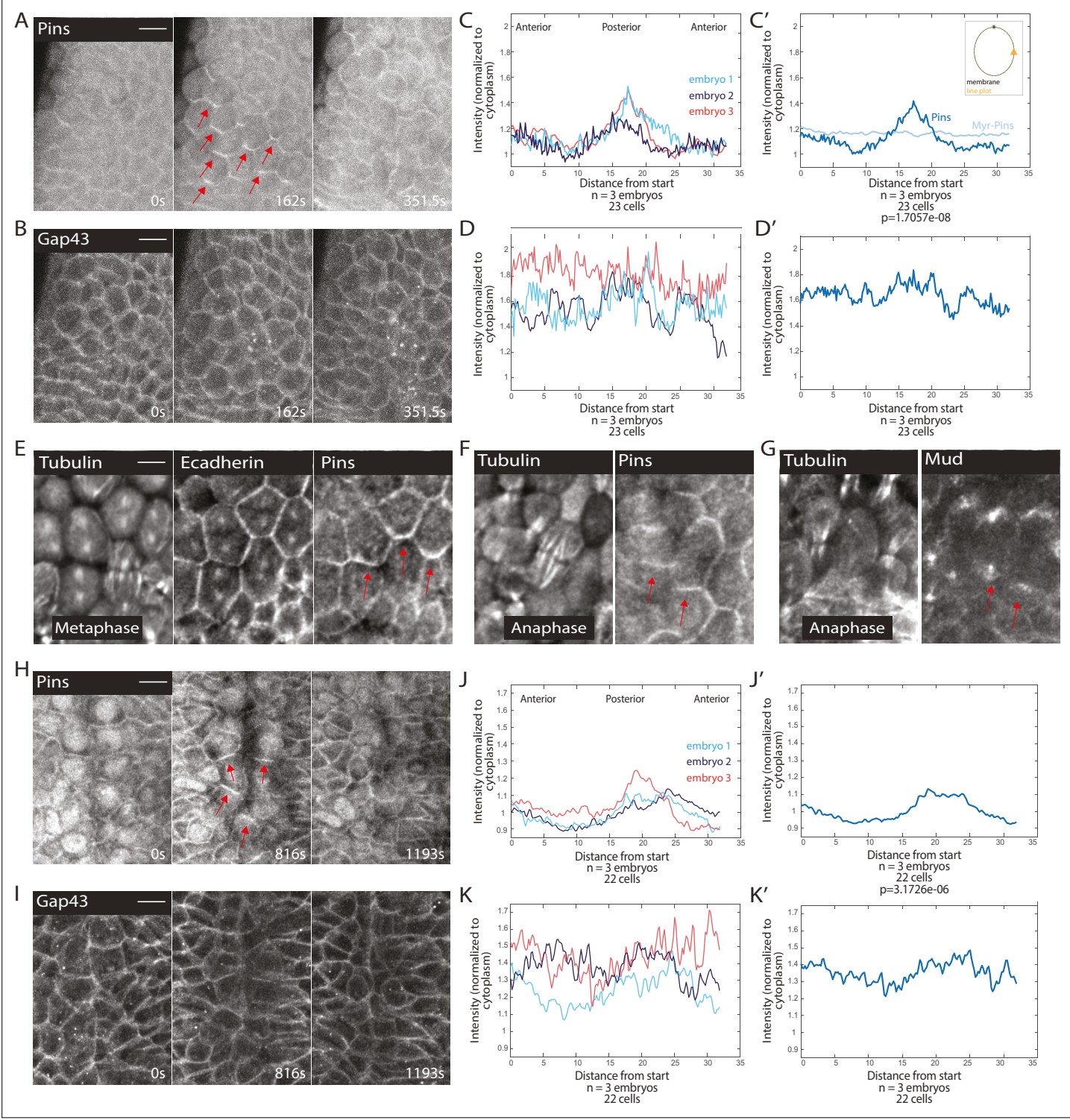

**Figure 2.** Pins is planar polarized in mitotic domain cells. (**A–B**) Time series of mitotic domain 1 (MD1) of embryo expressing Pins::YFP, to label spindle rotation machinery, and Gap43::mCherry, to mark membranes. scale bar is 10 μm. (**C, J**) Quantification of average cellular (7–9 cells) Pins::YFP signal intensity around the mitotic cell cortex for three separate embryos, lines indicate average of cells in each embryo for MD1 (**C**) or MD14 (**J**). Cortical intensity was measured at anaphase, indicated by cell elongation, and normalized to cytoplasmic Pins::YFP signal. (**C', J'**) Average of embryos in C or J. (**D, K**) Quantification of Gap43::mCherry (membrane marker) intensity for same cells and three embryos as in C for MD1 or J for MD14. (**D', K'**) Average of embryos in D, MD1, or K, MD14. Intensity was normalized to cytoplasmic Gap43::mCherry signal. (**E**) Fixed sample of metaphase cells in MD1 stained for tubulin (left), E-cadherin (center), and Pins (right). (**F**) Fixed sample of anaphase cells in MD1 stained for tubulin (left) and Pins (right). (**G**) Fixed sample

*Figure 2 continued on next page*

*Figure 2 continued*

of anaphase cells in MD1 stained for tubulin (left) and Mud (right). (**H–I**) Time series of MD14 cells expressing Pins::YFP and Gap43::mCherry, similar to A and B. scale bar is 10 μm.

The online version of this article includes the following video and figure supplement(s) for figure 2:

**Figure supplement 1.** Pins is planar polarized on the posterior side of mitotic domain 3 (MD3) and anterior side of MD5 cells.

**Figure 2—video 1.** Max intensity projection (with Gaussian blur) of control embryo expressing Pins::GFP cropped to focus on divisions of mitotic domain 1 (MD1).

https://elifesciences.org/articles/78779/figures#fig2video1

during mitosis and abolished Pins planar polarity (*Figure 2C'*, *Figure 3—figure supplement 2*). Importantly, perturbation using myr-Pins did not obviously disrupt other morphogenetic events during gastrulation. Mesoderm invagination, cephalic furrow invagination, and germ band extension all occur in embryos expressing myr-Pins. Further, expression of myr-Pins does not impact the number, timing, or position of cells dividing within the mitotic domains. We overexpressed the myristoylated version of Pins instead of using a Pins null mutant because we and others have previously shown that the Pins null mutant causes aberrant out-of-plane divisions and our goal was to disrupt Pins polarity while keeping the divisions in-plane (*Chanet et al., 2017*; *Izumi et al., 2004*). We have previously shown myristoylated Pins is functional and able to rescue off-axis divisions that occur after Dlg-RNAi (*Chanet et al., 2017*). Despite occurring in-plane, spindles in embryos expressing myristoylated Pins fail to orient along the AP axis. Instead, division angles are evenly distributed, occurring with seemingly no preference between the DV and AP axes for cells within MD1 (*Figure 3I–J*, *Figure 3—video 2*). In comparison, spindles in embryos expressing control Pins orient along the AP axis, with most divisions occurring within 30° of the AP axis (*Figure 3G–H*). Myr-Pins expressing embryos had an average division angle within MD1 that was significantly different from controls (42.7 and 20.1, respectively, p=0.000103). Results for MDs 3 and 5 are consistent with MD1 (*Figure 3—figure supplement 3*). Myristoylated Pins expression also disrupted division orientation in MD14. Myr-Pins expressing embryos had an average division angle within MD14 that was significantly different from controls (41.2 and 18.1, respectively, p=0.0079) (*Figure 3K–N*). These data suggest that Pins polarity is necessary for mitotic spindle orientation and planar polarity of divisions within MDs 1, 3, 5, and 14. Next, we investigated what mechanisms establish Pins planar polarity.

## Adherens junctions, but not Toll-like receptors or other PCP pathways, are required for division orientation

We hypothesized that Pins polarity could require an upstream planar cell polarity pathway and/or a mechanical cue. We first tested the requirement of planar cell polarity pathways. Some well-known planar polarity pathways include the Fat-Dachsous, the core PCP, and the PAR pathways (*Hale and Strutt, 2015*; *Strutt and Strutt, 2009*). The Fat-Dachsous and core PCP pathways were not good candidates for orienting divisions within MDs 1, 3, and, 5 because previous work has shown the components of these pathways are not expressed early enough and/or in the MDs to be involved (*Clark et al., 1995*; *Tomancak et al., 2002*; *Tomancak et al., 2007*). The PAR pathway was also not a good candidate because the *Drosophila* homologue of Par 3, Bazooka (Baz), did not exhibit planar cell polarity in the MDs, despite retaining apical localization during mitosis (*Ko et al., 2020*). Therefore, we explored the possible involvement of Toll-like receptors (TLRs), which act downstream of even-skipped (Eve) and have been shown to be necessary for the planar polarity of non-muscle Myo II and Baz in the *Drosophila* trunk (*Lavalou et al., 2021*; *Paré et al., 2014*). We investigated whether TLRs were involved in division orientation by injecting live embryos with dsRNA of TLRs 2, 6, and 8 and control 0.1× TE buffer. Furthermore, we analyzed fixed embryos that were triple mutants for these TLRs. Injection of dsRNA of TLRs 2, 6, and 8 did not have a significant impact on division orientation (*Figure 4—figure supplement 1A-D*). The triple TLR mutant perturbation was significantly different from controls but did not randomize division orientation to the extent of expressing Myr-Pins (*Figure 4—figure supplement 1E-F*). Finally, Myo II and Baz were not planar polarized in the *Drosophila* head (*Chanet et al., 2017*; *Ko et al., 2020*), leading us to conclude that TLRs are not solely responsible for division orientation in MDs 1, 3, 5.

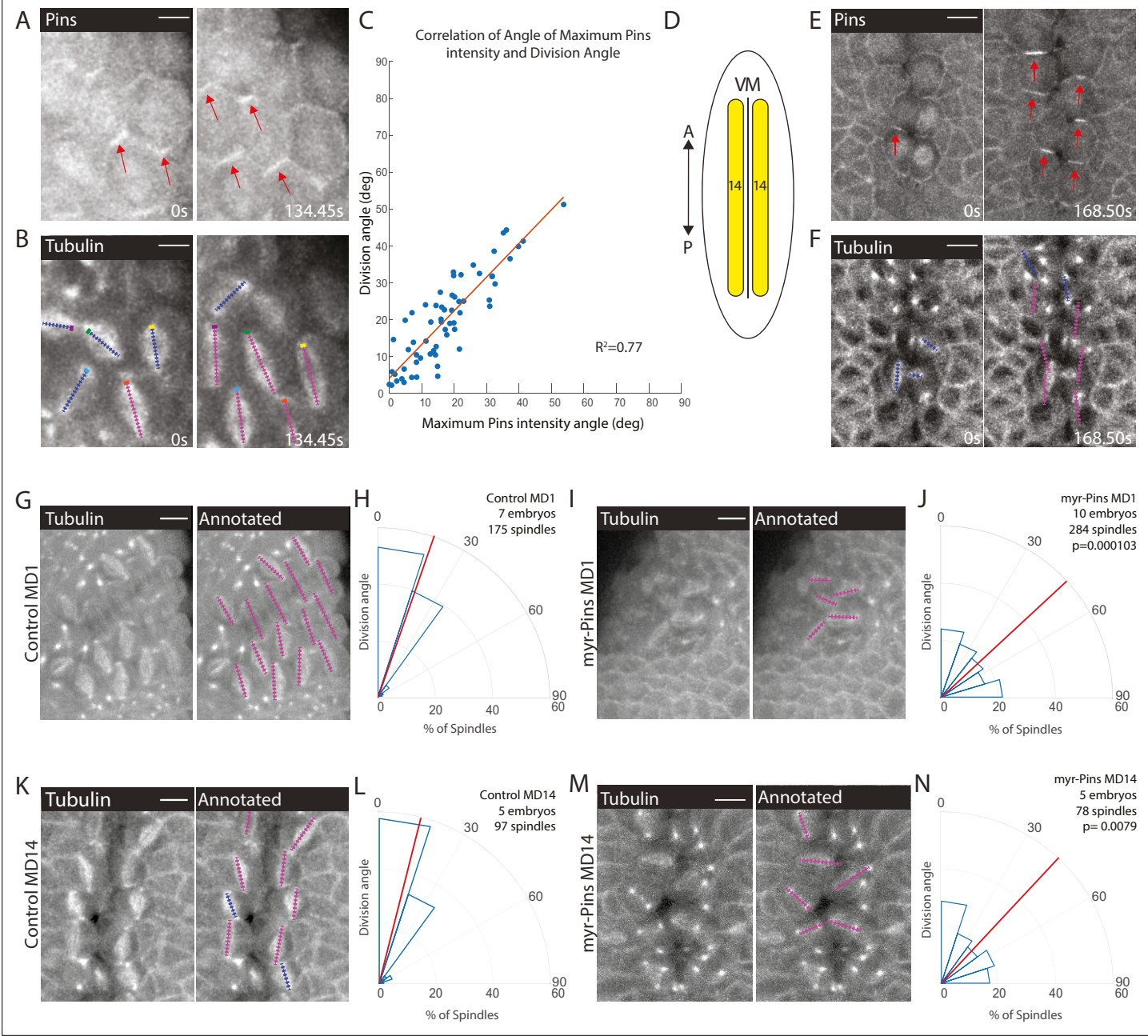

**Figure 3.** Planar polarized divisions depend on asymmetric Pins localization. (**A–B**) Pins localizes in planar polarized crescents immediately prior to spindle orientation. Images are time series of mitotic domain 1 (MD1) of embryo with labeled Pins (Pins::YFP) and labeled tubulin (Tubulin::mCherry), to mark mitotic spindles, blue lines indicate spindles that are still rotating, magenta lines indicate spindles that have reached their final orientation. Colored boxes at tips of lines label spindles with the color marking the same spindle at each time point. Scale bar is 10 µm. (**C**) Spindle orientation is correlated with Pins polarity. Plot of the correlation between the angle of maximum Pins::YFP intensity and division angle. $R^2$=0.77. (**D**) Schematic of MD14, located on either side of the ventral midline (VM). (**E–F**) Images are a time series of MD14 cells labeled with Pins::YFP and tubulin::mCherry, to mark mitotic spindles. (**G, I**) Myristoylated Pins disrupts division orientation. Left, image of MD1 of control embryo (G, UAS >PinsGFP) or MD1 of Myristoylated Pins expressing embryo (I, UAS >myr::Pins::GFP). Images show marker (CLIP170::GFP) to mark mitotic spindles (grayscale), scale bar is 10 µm. Right, same image annotated with dashed lines to indicate spindle position. (**H, J**) Quantification of spindle angles for control embryos (H, UAS >PinsGFP) using rose plots or Myristoylated Pins expressing embryos (J, UAS >myr::Pins::GFP). For control embryos (H, UAS-PinsGFP) MD1, n=7 embryos and 175 spindles. For Myristoylated Pins expressing embryos, n=10 embryos and 284 spindles. Average orientation angles per embryo were statistically different between control and Myristoylated Pins expressing embryos, Mann-Whitney, p=0.000103. (**K, M**) Pins asymmetry is required for MD14 spindle orientation. Left, image of MD14 of control embryo (G, UAS-PinsGFP) or MD14 of Myristoylated Pins expressing embryo (I, UAS >myr::Pins::GFP). Images show marker (CLIP170::GFP) to mark mitotic spindles (grayscale), scale bar is 10 µm. Right, same image annotated with

*Figure 3 continued on next page*

*Figure 3 continued*

dashed lines to indicate spindle position. (**L, N**) Quantification of spindle angles for control embryos (UAS >PinsGFP) or Myristoylated Pins expressing embryos (N, UAS >myr::Pins::GFP) using rose plots. For control embryos (L, UAS >PinsGFP) MD14, n=5 embryos and 97 spindles. For Myristoylated Pins expressing embryos, n=5 embryos and 78 spindles. Average orientation angles per embryo were statistically different between control and Myristoylated Pins expressing embryos, Mann-Whitney, p=0.0079.

The online version of this article includes the following video and figure supplement(s) for figure 3:

**Figure supplement 1.** Division angle is not correlated with interphase cell shape.

**Figure supplement 2.** Myristoylated Pins is uniformly recruited to the apical membrane.

**Figure supplement 3.** Results from mitotic domains (MDs) 3 and 5 are consistent with results from MD1.

**Figure 3—video 1.** Max intensity projection (with Gaussian blur) of control embryo expressing Pins::GFP and Tubulin::Ch (to label mitotic spindle) cropped to focus on divisions of mitotic domain 1 (MD1).
https://elifesciences.org/articles/78779/figures#fig3video1

**Figure 3—video 2.** Max intensity projection (with Gaussian blur) of control embryo (Rh3 RNAi) (left), embryo expressing α-catenin RNAi (center), or embryo expressing Myr-Pins (right) with fluorescently labeled Clip170::GFP (to label mitotic spindle) and cropped to focus on divisions of MD1.
https://elifesciences.org/articles/78779/figures#fig3video2

We next tested the function of AJs. AJs mechanically couple epithelial cells, serve as platforms for protein assembly, and are composed of cadherin-catenin complexes (*Hinck et al., 1994*; *Kovacs and Yap, 2008*; *Leckband and Prakasam, 2006*; *Oda et al., 1994*; *Rimm et al., 1995*; *Takeichi, 1991*; *Yap et al., 1997*). AJs are connected to F-actin, which is required for force transmission (*Desai et al., 2013*; *Rimm et al., 1995*; *Yonemura et al., 2010*). Junctional proteins, such as Canoe/Afadin (cno), have been implicated in spindle positioning (*Carminati et al., 2016*; *Speicher et al., 2008*; *Wee et al., 2011*). Previous work has shown that E-cadherin is able to recruit Pins in a tension-dependent manner (*Hart et al., 2017*). While AJs are not required for in-plane divisions in certain tissues (*Bergstralh et al., 2013*), they could serve as platforms for Pins to cue division orientation. To perturb AJs, we maternally depleted α-catenin (αcat) using RNAi. We have shown that αcat depletion is the most severe way to disrupt adherens junctions, totally uncoupling mesoderm cell actomyosin from the junctions, preventing their apical constriction, and lowering adherens junction levels throughout the entire embryo (*Denk-Lobnig et al., 2021*; *Yevick et al., 2019*). We crossed the RNAi (hairpin) stock to flies expressing a maternal GAL4 and fluorescently labeled tubulin to visualize spindles. When compared to a control RNAi (Rh3), embryos expressing the αcat RNAi had an even distribution of division angles in MD1, occurring with seemingly no preference between the DV and AP axes. αcat-depleted embryos exhibited an average division angle that was significantly different from controls (36.7 and 26.4, respectively; p=0.000583) (*Figure 4A–D*, *Figure 3—video 2*). αcat RNAi also disrupted spindle orientations in MD3 and MD5 (*Figure 4—figure supplement 2*). Perturbation using αcat RNAi does not recapitulate the complete disruption of division orientation seen in embryos expressing myr-Pins. This is likely due to the inherent variability in the RNAi knockdown. We observed embryos that had severe disruption of division orientation (*Figure 4E*) and embryos that had minor disruption and appeared more similar to controls (*Figure 4F*). We found that embryos with greater reductions in adhesion (i.e. more rounded cell morphology) exhibited more severe disruption of division orientation (*Figure 4E*), consistent with AJs and possibly force transmission being required for spindle orientation.

Because Cno has been shown to interact with Pins (*Speicher et al., 2008*; *Wee et al., 2011*), we tested Cno function in MDs 1, 3, and 5. To avoid the more catastrophic defects of a *cno* null mutant (*Sawyer et al., 2009*), we used two RNAi lines to maternally deplete Cno and crossed each to flies expressing fluorescently labeled tubulin. Measuring division angles, as previously described, we found that Cno depletion does not significantly disrupt division orientation within MDs 1, 3, and 5 compared to controls, Cno-depleted embryos exhibited an average division angle that was not significantly different from controls (27.9 and 26.4, respectively; p=0.2698) (*Figure 4G–H*). Note that Cno-RNAi with these lines disrupted myosin localization, similarly to genetic mutants (*Sawyer et al., 2011*; *Sawyer et al., 2009*), but did not fully abrogate midline convergence during ventral furrow formation (*Jodoin et al., 2015*), unlike αcat RNAi (*Denk-Lobnig et al., 2021*). Together these data suggest a role for AJs and possibly force transmission in spindle orientation. Given our results, we next tested whether AJs were required for polarization of Pins within MDs 1, 3, and 5.

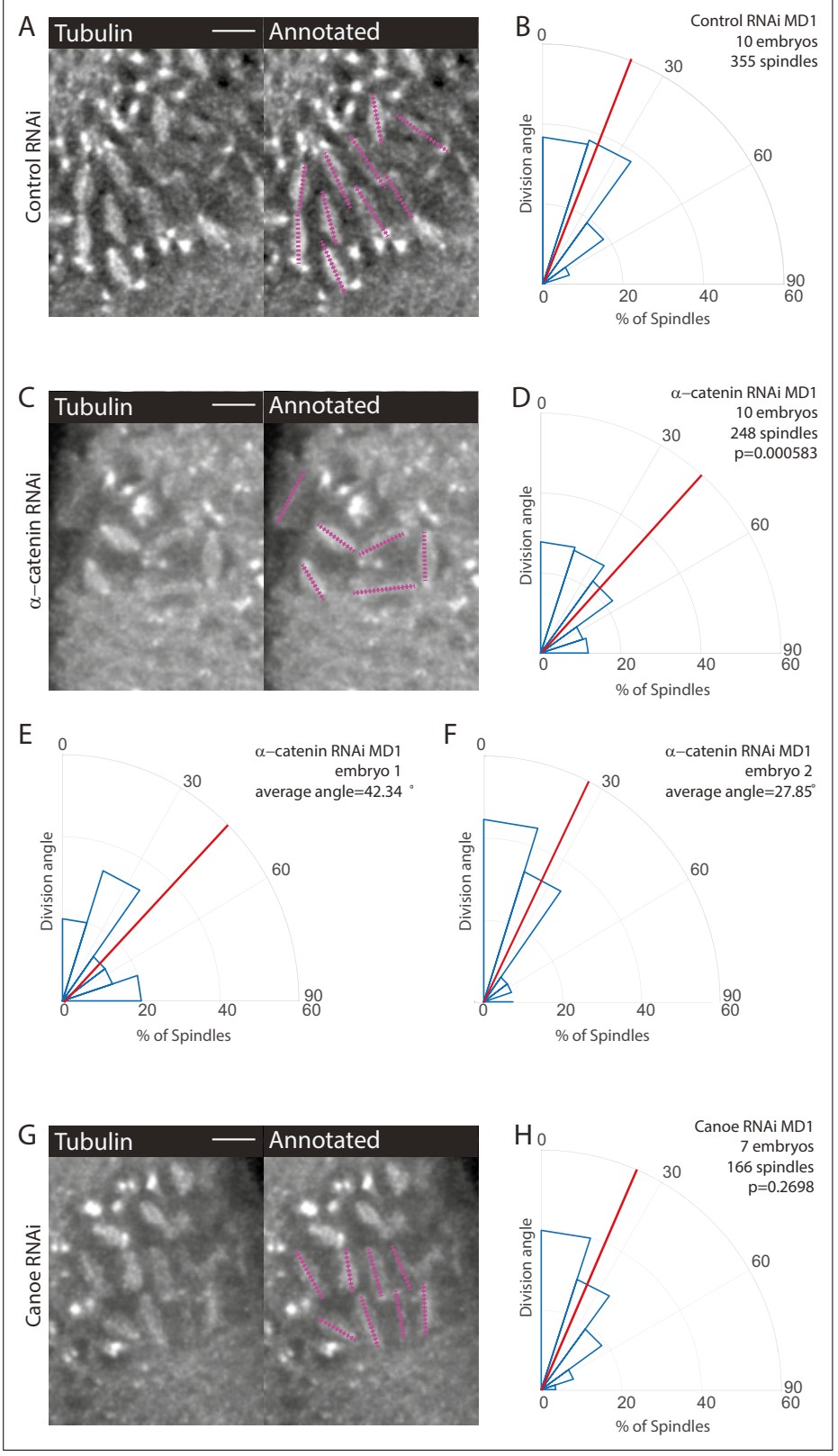

**Figure 4.** Disruption of adherens junctions, but not Canoe, inhibits planar polarized division orientation. (**A, C, G**) α-Catenin depletion disrupts division orientation. Left, image of MD1 of control knockdown (A, Rh3 RNAi), MD1 of α-catenin-depleted embryo (C, α-catenin RNAi), or MD1 of Canoe-depleted embryo (G, Canoe RNAi). Images show marker (CLIP170::GFP) to mark mitotic spindles (grayscale), scale bar is 10 µm. Right, same image annotated

*Figure 4 continued on next page*

*Figure 4 continued*

with dashed lines to indicate spindle position. (**B, D, H**) Quantification of spindle angles for control embryos (B, Rh3 RNAi), α-catenin-depleted embryos (D, α-catenin RNAi), or Canoe-depleted embryos (H, Canoe RNAi) using rose plots. For control embryos (Rh3 RNAi) MD1, n=10 embryos and 355 spindles. For α-catenin RNAi MD1, n=10 embryos and 248 spindles. For Canoe RNAi MD1, n=7 embryos and 166 spindles. Average orientation angles per embryo were statistically different between control and α-catenin RNAi, Mann-Whitney, p=0.000583, but not for canoe RNAi, Mann-Whitney, p=0.2698. (**E–F**) Quantification of spindle angles for individual α-catenin-depleted embryos using rose plots.

The online version of this article includes the following figure supplement(s) for figure 4:

**Figure supplement 1.** Disruption of Toll-like receptors (TLRs) does not disrupt polarized division orientation.

**Figure supplement 2.** Results from mitotic domains (MDs) 3 and 5 are consistent with results from MD1.

## Disruption of adherens junctions disrupts Pins polarization

To determine whether AJs are involved in Pins planar polarized localization, we examined fluorescently labeled Pins in αcat-depleted embryos. In contrast to control embryos, we found that Pins planar polarity is abolished when αcat was depleted (*Figure 5A–D*, *Figure 5—video 1*). Pins appeared to have no preferential localization, though given that cells divide in the epithelial plane, a process that depends on Pins, Pins must still localize laterally. Pins peak-to-trough ratio at the posterior was different in controls compared to αcat-depleted embryos (1.33±0.0751 and 1.06±0.0173, respectively, mean and SD, p=1.1887e-08). These data demonstrate a role for AJs in Pins planar polarization in MD1.

## Mechanical cues affect Pins polarization and division orientation

Given our results that AJs are involved in division orientation and Pins polarity and our finding that Pins is planar polarized within the force-dependent oriented divisions of MD14, we hypothesized that force transduction through AJs contributed to Pins polarization and division orientation in MDs 1, 3, and 5. To further test the role of force, we injected embryos expressing fluorescently labeled Pins and tubulin with a low dose of cytochalasin D (CytoD) that we have shown to inhibit force transmission without globally depolymerizing F-actin (*Jodoin et al., 2015*; *Mason et al., 2013*). Actomyosin is also necessary for cell rounding, which is necessary for planar divisions for MD cells (*Chanet et al., 2017*). As we were interested in the orientation of planar divisions, we focused our analysis on divisions that occurred within the plane of the epithelium. We found that injections of CytoD disrupted planar polarized division orientation, CytoD injected embryos exhibited an average division angle that was significantly different from controls (53.7 and 21.9, respectively; p=0.0043) (*Figure 6B, D, F and H*). CytoD injection also disrupted spindle orientations in MDs 3 and 5 (*Figure 6—figure supplement 1*). In addition, Pins localization was disrupted in embryos injected with CytoD whereas Pins exhibited normal polarity in DMSO injected controls (*Figure 6A, C, E and G*, *Figure 5—video 1*). The posterior Pins peak-to-trough ratio was different for DMSO injected controls and CytoD injected embryos (1.17±0.0685 and 1.08±0.0609, respectively, mean and SD, p=6.5007e-11). Interestingly, DMSO injection seemed to perturb the synchrony of divisions. In non-injected embryos, cells within the domain divide in a wave-like pattern, starting in the middle of the domain and propagating to the domain boundary (*Di Talia and Wieschaus, 2012*). In DMSO injected embryos, nearly all cells round and divide simultaneously. This division timing disruption did not disrupt division orientation or Pins polarization. As CytoD and αcat RNAi have many effects beyond disrupting forces, we next sought to test the role of tissue mechanics without chemical or genetic perturbation.

To that end, we performed laser ablation experiments to mechanically isolate MDs from the rest of the tissue. We focused on MD1 and performed a series of four cuts to isolate the domain (*Figure 6I*). We were unable to look at the effect on Pins polarization because the ablation bleached the Pins::YFP signal. Using embryos that expressed fluorescently labeled tubulin, we analyzed the effect of mechanical isolation on spindle orientation. Cells that divide within the mechanically isolated region fail to orient along the AP axis (*Figure 6K and M*). As an internal control, we utilized the mirrored spatial pattern of MD1 and cut one side of the domain and left the other side uncut (*Figure 6I*). In the same embryo, we were able to observe spindles within the cut region fail to orient and neighboring cells outside the cut region orient correctly (*Figure 6J-M*, *Figure 6—video 1*). As a further control, and

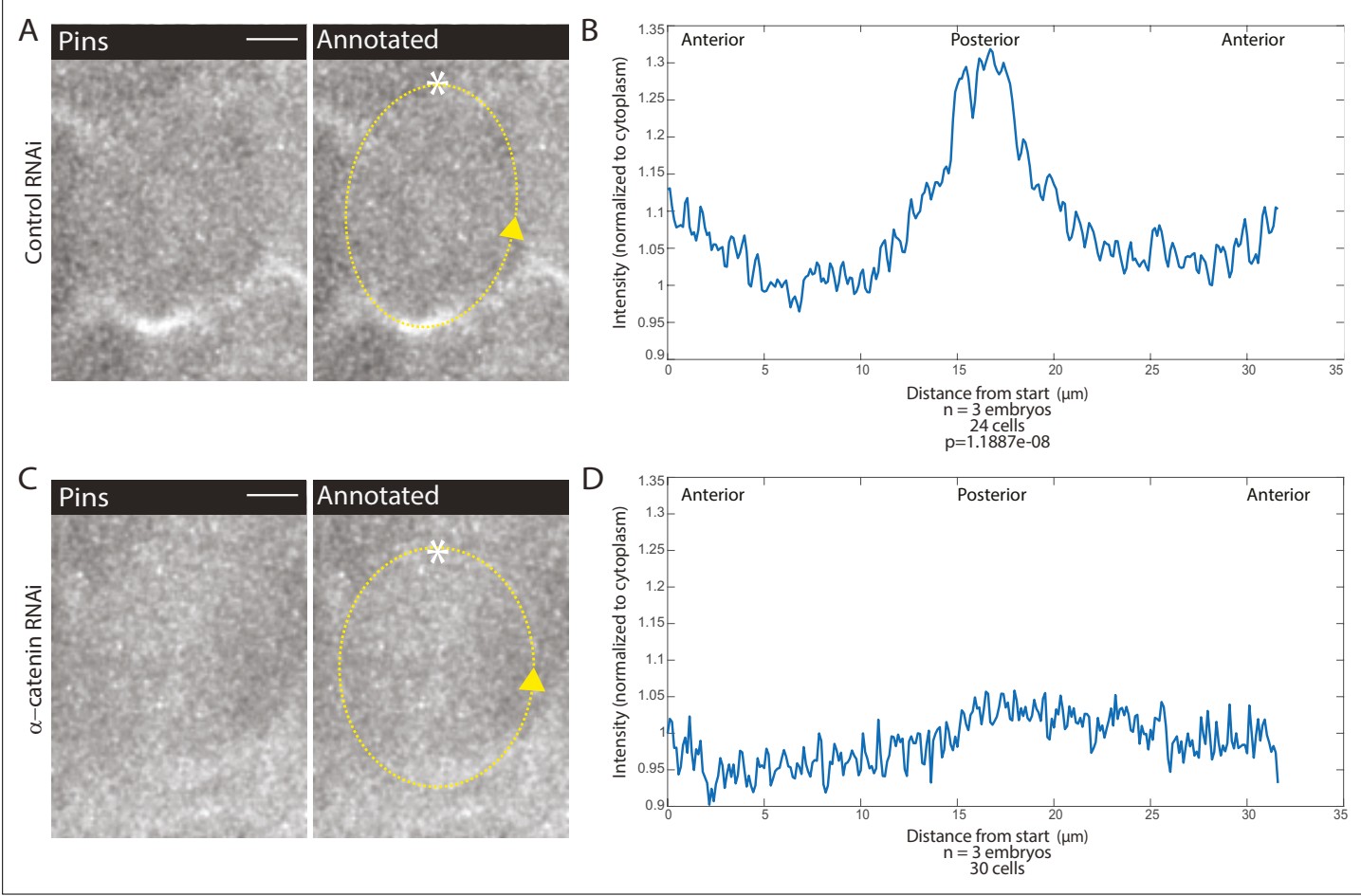

**Figure 5.** Adherens junction depletion disrupts Pins planar cell polarity. (**A, C**) αCatenin depletion disrupts Pins localization. Left, image of a mitotic domain 1 (MD1) cell of control embryo (A, Rh3 RNAi) or an MD1 cell of α-catenin-depleted embryo (C, α-catenin RNAi). Images show labeled Pins (Pins::YFP), scale bar is 10 μm. Right, same image annotated with dashed line to indicate cell outline used to plot intensity profile, * to indicate start, and arrow to indicate direction of line. (**B, D**) Quantification of Pins intensity for control embryos (Rh3 RNAi) or α-catenin-depleted embryos (D, α-catenin RNAi). Intensity was normalized to cytoplasm. For control (Rh3) MD1, n=3 embryos and 25 cells. For α-catenin RNAi MD1, n=3 and 30 cells.

The online version of this article includes the following video for figure 5:

**Figure 5—video 1.** Max intensity projection (with Gaussian blur) of control embryo (Rh3 RNAi) (left), embryo expressing α-catenin RNAi (center), or embryo injected with 0.25 mg/mL cytochalasin D (cytoD) (right) with fluorescently labeled Pins::YFP and cropped to focus on divisions of mitotic domain 1 (MD1).

https://elifesciences.org/articles/78779/figures#fig5video1

to compensate for the difficulty in acquiring images that permitted visualization of both sides of the domain, we performed sham ablations, where we irradiated the region with a level of light below the cutting threshold. In sham ablated control embryos, spindles oriented with preference toward the AP axis and had an average division angle that was significantly different from cells within the cut region (20.0 and 58.0, respectively; p=0.0286) (*Figure 6J-M*, *Figure 6—figure supplement 2E*).

To test whether forces transmitted from either cephalic furrow invagination (posterior to MDs) or ventral furrow (ventral to MDs) – mesoderm invagination – are required for oriented divisions, we made directional incisions to separate MD1 from these morphogenetic movements. The cephalic furrow is closest to MDs 1, 3, and 5 and could be expected to pull the MD cells along the AP axis. We made DV oriented (horizontal) laser ablations in the posterior part of MD5 to separate the dorsal head from the cephalic furrow and found that cells exhibited oriented divisions, occurring with an average division angle similar to control sham ablations (18.3 and 20.0, respectively; p=1) (*Figure 6N*, *Figure 6—figure supplement 2C D E*, *Figure 6—video 2*). In contrast, we found that AP oriented line ablations ventral to the head MDs, which separated them from putative pulling forces from the mesoderm,

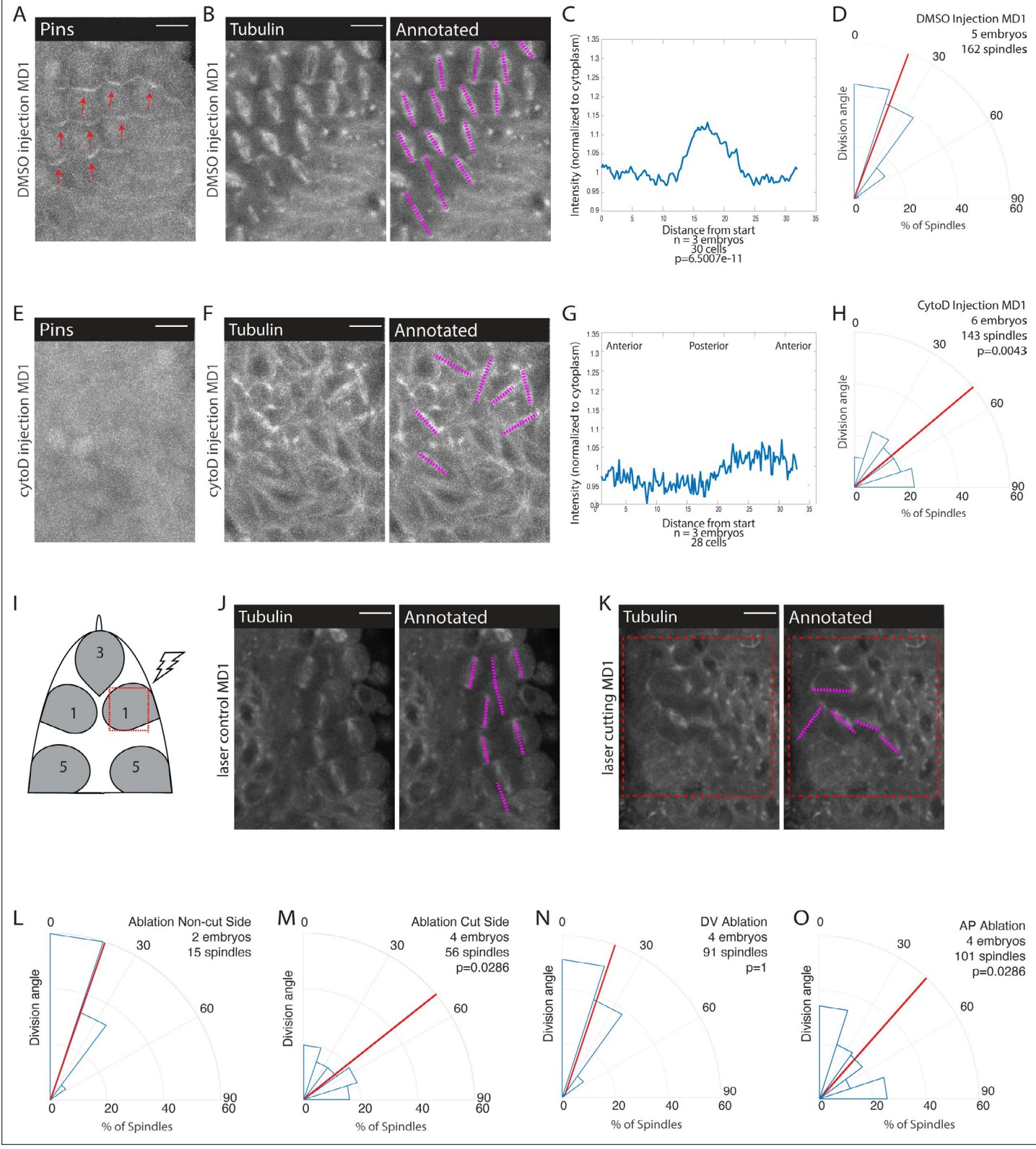

**Figure 6.** Mechanical force is necessary for Pins polarity and spindle orientation. (**A, E**) Cytoskeletal disruption prevents Pins localization. Image of mitotic domain 1 (MD1) in control embryo (A, DMSO injection) or MD1 of cytochalasin D (CytoD) injected embryo (E, 0.25 mg/mL CytoD injection). Images show labeled Pins (Pins::YFP), red arrows point to crescents of Pins in control MD1, scale bar is 10 μm. (**B, F**) Cytoskeletal disruption interferes with division orientation. Image of MD1 of control embryo, same embryo as A, (B, DMSO injection) or MD1 of CytoD injected embryo (**F**). Images, from

*Figure 6 continued on next page*

*Figure 6 continued*

same embryo as E, show marker (Tubulin::mCherry) to mark mitotic spindles (grayscale), scale bar is 10 µm. Right, same image annotated with dashed lines to indicate spindle position. (C, G) Quantification of Pins intensity for control embryos (C, DMSO injection) or CytoD injected embryo (G, 0.25 mg/mL CytoD injection). Intensity was normalized to cytoplasm. For control (DMSO injection) MD1, n=3 embryos and 30 cells. For CytoD injection MD1, n=3 and 28 cells. (D, H) Quantification of spindle angles for control embryos (D, DMSO injection) or CytoD injected embryo (H) using rose plots. For control embryos (DMSO injection) MD1, n=5 embryos and 162 spindles. For CytoD injection MD1, n=6 embryos and 143 spindles. When compared to control (DMSO injection) using Mann-Whitney, p=0.0043. (I) Schematic of laser ablation schema. The ROI for laser cutting was drawn around MD1 on one side of the embryo to mechanically uncouple MD1 from surrounding tissue. The other side of MD1 was not cut. (J–K) Separating MD1 from surrounding tissue disrupts spindle orientation. Left, image of MD1 on control side of embryo (J, non-cut) or laser cut side of embryo (K). Images show marker (Tubulin::mCherry) to mark mitotic spindles (grayscale), scale bar is 10 µm. Right, same image annotated with dashed lines to indicate spindle position, red box indicates laser cut ROI. (L-M) Quantification of spindle angles for MD1 of control side (L, non-cut), or laser cut side (M) using rose plots. For control side (non-cut) MD1, n=2 embryos and 15 spindles. For laser cut side MD1, n=4 embryos and 56 spindles. Average orientation angles per embryo were statistically different between laser cut embryos and sham ablation embryos, Mann-Whitney, p=0.0286. (N-O) Quantification of spindle angles for dorsal-ventral (DV) cut embryos (N) or anterior-posterior (AP) cut embryos (O) using rose plots. For DV cut embryos MD1, n=4 embryos and 91 spindles. Average orientation angles per embryo were not different between DV cut and sham ablated control embryos, Mann-Whitney, p=1. For AP cut embryos MD1, n=4 embryos and 101 spindles. Average orientation angles per embryo were different between AP cut and sham ablated control embryos, Mann-Whitney, p=0.0286.

The online version of this article includes the following video and figure supplement(s) for figure 6:

**Figure supplement 1.** Results from mitotic domains (MDs) 3 and 5 are consistent with results from MD1.

**Figure supplement 2.** Laser cutting schema and control.

**Figure 6—video 1.** Max intensity projection (with Gaussian blur) of embryo expressing Tubulin::Ch.
https://elifesciences.org/articles/78779/figures#fig6video1

**Figure 6—video 2.** Max intensity projection (with Gaussian blur) of embryos expressing Tubulin::Ch.
https://elifesciences.org/articles/78779/figures#fig6video2

resulted in a random distribution of division angles, similar to complete isolation of the MD (40.0 and 20.0, respectively; p=0.0286) (*Figure 6O*, *Figure 6—figure supplement 2A B E*, *Figure 6—video 2*).

## Mesoderm invagination polarizes Pins and division orientation in the embryonic head

The divisions in MDs 1, 3, and 5 occur during gastrulation, ventral furrow formation pulls ectoderm cells around the circumference of the embryo (*Lye et al., 2015*; *Rauzi et al., 2015*), and separation of the MDs from the ventral region disrupts division orientation. Therefore, we hypothesized that forces from mesoderm invagination, which occurs ~10 min before division, could cue Pins polarization and division orientation. To perturb mesoderm invagination, we disrupted the transcription factor Snail because Snail depletion most severely depletes apical actomyosin activity, and thus pulling forces, in the ventral furrow (*Martin et al., 2009*; *Mason et al., 2013*). We injected Snail dsRNA into embryos expressing fluorescently labeled tubulin and Pins. We found that Snail depletion disrupted division orientation (*Figure 7B, D, F and H*, *Figure 7—video 1*). Embryos injected with Snail dsRNA had a random distribution of division angles, occurring with an average division angle significantly different from controls (40.0 and 22.6, respectively; p=0.0121). In addition, Pins polarity was clearly disrupted in embryos injected with Snail dsRNA. For 0.1× TE buffer injected controls, the posterior Pins peak-to-trough ratio was different from Snail dsRNA injected embryos (1.27±0.08401 and to 1.10±0.10545, respectively, mean and SD, p=6.7956e-08) (*Figure 7A, C, E and G*). Beyond division orientation and Pins polarity, injections of Snail dsRNA disrupted the patterning of MDs 1, 3, and 5. Instead of three distinct domains, a single group of anterior cells divide. Taken together, these data indicate that morphogenetic forces play a role in Pins polarity and division orientation within MDs 1, 3, and 5. A role for morphogenetic forces is consistent with our observation that disruption of αcat, actomyosin (with CytoD), or laser ablation disrupt division orientation.

## Discussion

Here, we discovered that Pins exhibits planar cell polarity in mitotic domains of the dorsal head (MDs 1, 3, and 5) and along the ventral midline (MD14). We demonstrated that Pins polarity is required for spindle orientation by overexpressing a myristoylated version of Pins that exhibits unregulated

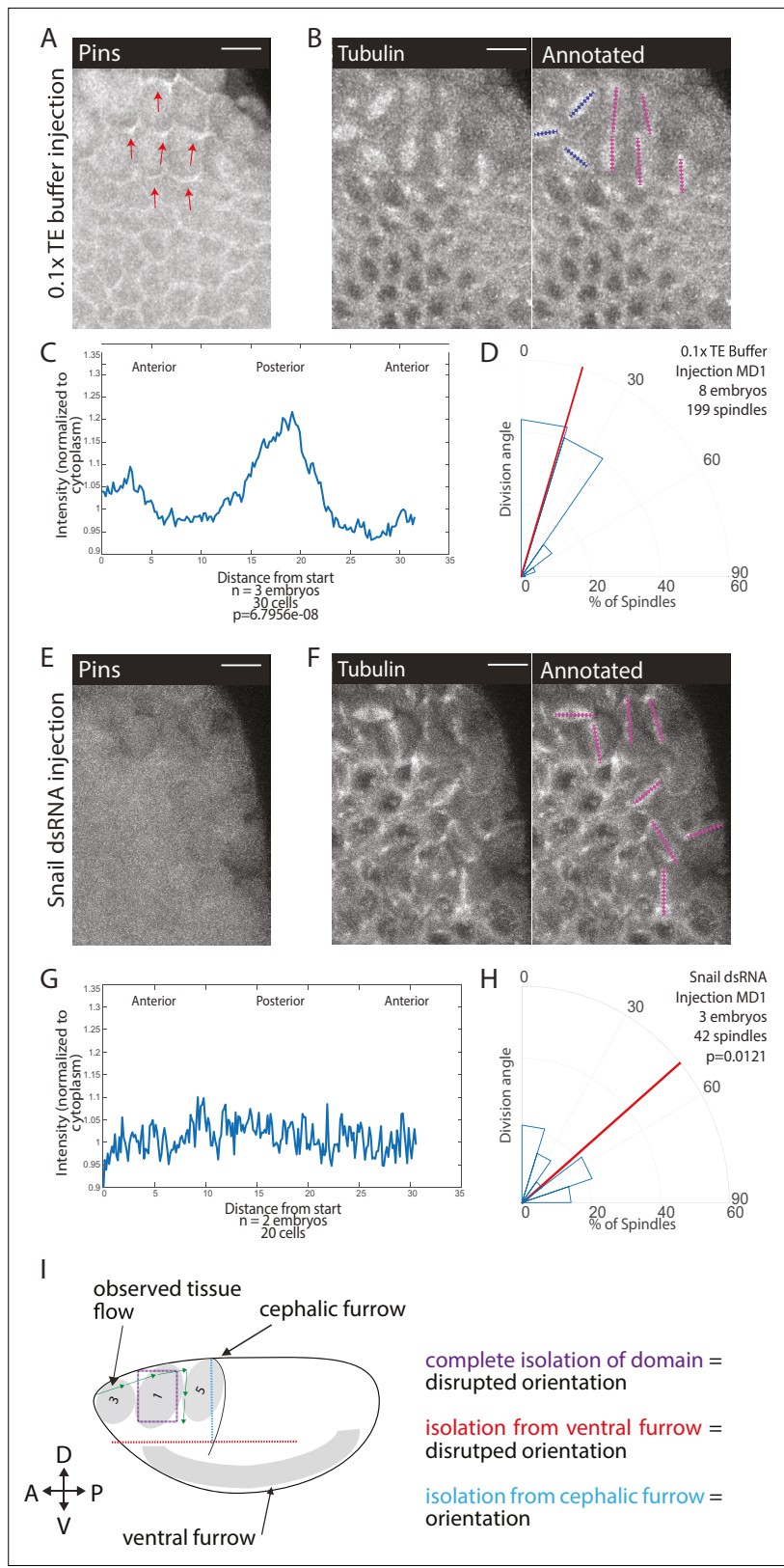

**Figure 7.** Snail depletion disrupts Pins polarization and spindle orientation.
(**A,E**) Disrupting ventral furrow formation disrupts Pins polarity. Image of mitotic domain 1 (MD1) of control embryo (A, 0.1× TE buffer injection) or MD1 of Snail dsRNA injected embryo (**E**). Images show labeled Pins (Pins::YFP), red arrows point to crescents of Pins in control, scale bar is 10 µm. (**B, F**) Snail depletion disrupts

*Figure 7 continued on next page*

*Figure 7 continued*

division orientation. Left, image of MD1 of control embryo (B, 0.1× TE buffer injection) or MD1 of Snail-depleted embryo (**F**). Images show marker (Tubulin::mCherry) to mark mitotic spindles (grayscale), scale bar is 10 µm. Right, same image annotated with dashed lines to indicate spindle position. (**C, G**) Quantification of Pins intensity for control embryos (C, 0.1× TE buffer injection) or Snail dsRNA injected embryo (**G**). Intensity was normalized to cytoplasm. For control (0.1× TE buffer injection) MD1, n=3 embryos and 30 cells. For Snail depletion MD1, n=2 and 20 cells. (**D, H**) Quantification of spindle angles for control embryos (H, 0.1× TE buffer injection) or Snail dsRNA injected embryo (**H**) using rose plots. For control embryos (0.1× TE buffer injection) MD1, n=8 embryos and 199 spindles. For Snail dsRNA injection MD1, n=3 embryos and 42 spindles. Average orientation angles per embryo were statistically different between control and Snail dsRNA injected embryos, Mann-Whitney, p=0.0121. (**I**) Schematic summary of laser cutting experiments. Complete domain isolation results in disrupted division orientation. Isolation from the ventral furrow through anterior-posterior (AP) cutting also resulted in disrupted division orientation, whereas isolation from the cephalic furrow through dorsal-ventral (DV) cutting did not affect division orientation.

The online version of this article includes the following video for figure 7:

**Figure 7—video 1.** Top. Max intensity projection (with Gaussian blur) of control embryo (0.1× TE buffer injection) (left) or embryo injected with Snail dsRNA (right) with fluorescently labeled Pins::YFP and cropped to focus on divisions of mitotic domain 1 (MD1).

https://elifesciences.org/articles/78779/figures#fig7video1

---

cortical localization and disrupts spindle orientation. We found that Pins planar cell polarity and AP spindle orientation depend on adherens junctions. This and our finding that actomyosin disruption also prevents Pins planar cell polarity led us to hypothesize that force transmission through the tissue orients Pins and cell divisions. We showed that forces from ventral furrow formation were critical for spindle orientation, by (1) mechanically isolating MD1 cells from all surrounding tissue (*Figure 7I*, purple lines), (2) mechanically isolating MD1 cells specifically from the ventral furrow via directional laser ablations (*Figure 7I*, red lines), and (3) showing that disrupting ventral furrow formation via Snail depletion abrogates proper spindle orientation and Pins localization.

## Other mechanisms of division orientation

Several other mechanisms of planar polarized division orientation have been identified. Previous works have shown that planar cell polarity pathways, namely the Fat/Dachsous, the core PCP, and the PAR pathways, coordinate divisions of the fly wing disc, during zebrafish gastrulation, and in *Drosophila* SOP cells, among other contexts (*Baena-López et al., 2005*; *Besson et al., 2015*; *Concha and Adams, 1998*; *Gho and Schweisguth, 1998*; *Gong et al., 2004*; *Mao et al., 2006*). We ruled out the Fat/Dachsous and core PCP pathways because previous studies indicated that the components of these pathways are not expressed early enough in development to play a role in orienting the divisions of the dorsal head or ventral midline (*Clark et al., 1995*; *Tomancak et al., 2002*; *Tomancak et al., 2007*). We further confirmed this by examining fluorescently tagged versions of proteins within these pathways. As was reported, we found no evidence of appreciable expression of these proteins (data not shown). Further, we ruled out the PAR pathway because we saw no evidence of planar polarity in Bazooka/Par 3 (*Ko et al., 2020*). The TLR planar polarity pathway that polarizes myosin in the *Drosophila* trunk was a candidate for involvement in the mitotic domains (*Paré et al., 2014*). We ruled out this pathway for two reasons, first the TLR pathway in the *Drosophila* trunk results in polarized Myosin and Bazooka/Par3. Observation of these proteins in the dorsal head showed no strong polarization of either protein (*Chanet et al., 2017*; *Ko et al., 2020*). Second, we generated a triple mutant of the TLRs identified and performed injections of dsRNA of these TLRs. In each case, spindle orientation was not affected to the extent of the myristoylated Pins perturbation, indicating TLRs cannot be solely responsible for spindle orientation in MDs 1, 3, and 5.

Hertwig's rule or the long axis rule is perhaps one of the most established mechanisms of division orientation. More recent studies have shown that tricellular junctions are predictors of division orientation. Mud localizes to tricellular junctions which cues spindle orientation (*Bosveld et al., 2016*). We did not observe Mud localization to tricellular junctions within the dorsal head mitotic domains and along the ventral midline. Furthermore, we find no correlation between cell long axis and division orientation within these domains, while we find a strong correlation between Pins polarity and cell division angle.

## Myosin-dependent tension and division orientation

Myosin-dependent tension has been shown to cue the orientation of the mitotic spindle. A growing body of work suggests a role for tension in division orientation, with notable examples in fly development. In the fly notum, it was shown that cells rely on isotropic myosin-dependent tension to sense cell shape and divide along the cell long axis (*Lam et al., 2020*). Unlike the cells of the fly notum, cells within MDs 1, 3, 5, and 14 do not follow the long axis rule. However, isotropic myosin-dependent tension is required for cell rounding and for symmetric cell divisions in MD cells (*Chanet et al., 2017*).

Tension has also been shown to overrule the long axis rule and orient divisions with the axis of tension rather than the cell long axis. In the follicular epithelium, cortical tension at the apical surface cues planar divisions in a Canoe-dependent manner (*Finegan et al., 2019*). In this case, Pins and Mud were not planar polarized but rather localized around the entire cell cortex and did not play a role in the planar divisions within the follicular epithelium. Similarly, in parasegmental boundaries, local anisotropy in myosin-dependent tension overrides cell shape. In this context it was shown that tension is both necessary and sufficient to cue division orientation (*Scarpa et al., 2018*). Pins and Mud were shown to not be involved in division orientation but were necessary for planar divisions. In the head MDs, we did not observe anisotropic myosin-dependent tension. Instead, we observe a uniform distribution of actin and myosin along interfaces of cells. While myosin-dependent tension plays an important role in division orientation in different contexts within fly development, we found no evidence for anisotropic myosin orienting dorsal head divisions.

## Adherens junctions and division orientation

Prior work suggested that adherens junctions are involved in division orientation. Notably, within *Drosophila* male germline stem cells (*Inaba et al., 2010*) and SOP cells (*Le Borgne et al., 2002*). The adherens junction component, E-cadherin, is localized asymmetrically and coordinates the asymmetric divisions within these cells and perturbation of E-cadherin results in misorientation of the mitotic spindle. Interestingly, in SOP cells it was also shown that perturbation of E-cadherin disrupted the Pins localization, although Pins still appears as a crescent (*Le Borgne et al., 2002*).

E-cadherin also has a demonstrated role in symmetric divisions. In MDCK cells E-cadherin recruits Pins in a tension-dependent manner (*Hart et al., 2017*). It was previously known that Pins/LGN localizes to junctions, but this study first showed a role for force in Pins' junctional localization (*Hart et al., 2017*). Our findings are consistent with these results. While we did not find a role for Cno in oriented cell division, we were unable to analyze division orientation in a *cno* null mutant, because these mutants also disrupt mesoderm invagination (*Sawyer et al., 2009*). As seen in the MDCK cells, in cells within the dorsal head mitotic domains and ventral midline, adherens junctions may sense anisotropic force or shear forces within the tissue and recruit spindle rotation machinery in a polarized fashion to cue division orientation. Mechanical tension and tissue flow has been shown to orient other planar cell polarity proteins (*Aigouy et al., 2010*; *Aw et al., 2016*). To our knowledge, this is the first in vivo example where mechanical force has been shown to polarize Pins to mediate division orientation.

## Role of divisions in morphogenesis

Whole-embryo light-sheet imaging has revealed a connection between mesoderm invagination and posterior, mediolateral flows in the head (*Stern et al., 2022*). These flows could explain how large-scale tissue movements on the ventral side of the embryo are connected to oriented division in the dorsal head. An important question that remains is the function of the oriented divisions of the dorsal head during morphogenesis. The oriented divisions of the ventral midline (MD14) have been shown to alleviate tissue strain and promote tissue elongation by promoting an increase in surface area of the mesectoderm (*Wang et al., 2017*). This agrees with other studies which suggest that oriented division facilitate tissue elongation during development (*Aigouy et al., 2010*; *Baena-López et al., 2005*; *Gong et al., 2004*; *Mao et al., 2011*). Perturbation using myr-Pins did not affect the number of dividing cells within the domains or the timing of onset of divisions but could possibly affect the elongation of the head epithelium. A recent study followed all the cell intercalations, movements, and divisions during gastrulation, showing cell divisions compensate for the cell loss due to various invaginations (*Stern et al., 2022*). Indeed, past work has shown that cell division can lead to oriented pushing forces that could contribute to morphogenesis (*Gupta et al., 2021*). In the embryo, apical relaxation that results from mitotic entry and the following oriented division has been shown to give

rise to ectopic tissue folds (*Ko et al., 2020*). Therefore, we speculate that the oriented division of the dorsal head facilitates tissue elongation and compensates for cell loss from mesoderm and endoderm invagination.

### Study limitations and future directions

While the evidence in this study supports the hypothesis that morphogenic forces polarize Pins which is required for division orientation, the interpretation of this evidence is limited by certain aspects of the study. First, in describing Pins crescents in mitotic cells, we argue that Pins is on the posterior side for MDs 1 and 3 and anterior side for MD5 (*Figure 2A and C*, *Figure 2—figure supplement 1*). This result is based on the observation of cells on the boundary of mitotic domains that are not surrounded by other mitotic cells. We do not have the resolution to determine whether Pins is on one or both sides of cells within the domains. One question that arises from this observation is: how does a junction that is under tension recruit Pins to just one side? It will be of future interest to understand the sidedness of Pins in the mitotic domains and what causes this sidedness.

A broader question is how Pins is specifically recruited along the AP axis. While we show that mesoderm invagination is necessary for division orientation, what provides the directionality of the divisions remains unknown. Previous work has shown tissue flows that correspond to the division orientations we observe (*Streichan et al., 2018*). More work is needed to determine the nature of the forces generated at the ventral midline that orient divisions in the dorsal head.

Finally, we can only speculate on the significance of these oriented divisions during morphogenesis. Planar, oriented division has been shown to be involved in tissue elongation and folding (*Baena-López et al., 2005*; *Kulukian and Fuchs, 2013*; *Lancaster and Knoblich, 2012*; *Morin and Bellaïche, 2011*). Given that the divisions of MDs 1, 3, 5, and 14 take place during gastrulation, we hypothesize that the divisions may aid in morphogenic movements such as germ-band extension and cephalic furrow invagination, but we have not directly seen this. These questions will form the basis for future work on the relationship between mechanical and molecular cues in division orientation and the contribution of division orientation during development.

## Materials and methods

### Fly stocks and genetics

Fly stocks and crosses used in this study are listed in *Supplementary file 1*. Crosses were maintained at 25°C. In the F1 generation, non-balancer females and males were used to set up cages that were incubated at 25°C. The F2 generation was used for imaging.

### Live and fixed imaging

For live imaging, embryos were dechorionated in 50% bleach for 2 min, washed in water, and mounted onto a glass slide coated with embryo glue (double-sided tape dissolved in heptane). No. 1.5 coverslips coated in glue were attached to the slide to use as spacers and a No. 1 coverslip was attached on top to create a chamber. Halocarbon 27 oil was used to fill the chamber. All imaging took place at 23–25°C.

For imaging fixed embryos, embryos were dechorionated in 50% bleach (Chlorox) for 2 min, washed in water, and then fixed in 4% paraformaldehyde (Electron Microscopy Sciences) in 0.1 M phosphate buffer at pH 7.4 with 50% heptane (Alfa Aesar) for 30 min and manually devitellinized (TLR mutant stains) or devitellinized by removing fixative, adding 50% methanol (Sigma Aldrich), and vortexing (Pins and Mud). Manually devitellinized embryos were stored in 0.01% Tween 20 in PBS (PBS-T and embryos devitellinized with 50% methanol were stored in 100% methanol at −20°C and rehydrated in PBS-T).

Embryos were washed in PBS-T, blocked with 10% BSA in PBS-T, and incubated with antibodies diluted in PBS-T. Embryos were incubated with primary antibodies at room temperature for 2 hr. Antibodies used in this study are listed in *Supplementary file 1*. Secondary antibodies used were conjugated with Alexa Fluor 488, 568, or 647 (Invitrogen) diluted at 1:500 in 5% BSA in PBS-T incubated overnight at 4°C. After antibody incubation, embryos were mounted onto glass slides using AquaPolymount (Polysciences).

All images were taken on a Zeiss LSM 710 confocal microscope with a 40×/1.2 Apochromat water objective lens, an argon-ion, 561 nm diode, 594 nm HeNe, 633 HeNe laser, and Zen software. The pinhole was set to 1 airy unit (au) for fixed imaging and 2.5 au for live imaging. For two-color live imaging, band-pass filters were set at ~490–565 nm for GFP and ~590–690 nm for mCherry (mCh). For three-color imaging, band-pass filters were set at ~480–560 nm for Alexa Fluor 488, ~580–635 nm for Alexa Fluor 568, and ~660–750 nm for Alexa Fluor 647.

## dsRNA injections

Primers included the sequence of the T7 promoter (TAATACGACTCACTATAGGGAGACCAC) followed by the following recognition sequences:

> Toll 2 F: AGTTTGAATCGAAACGCGAG
> Toll 2 R: GGACACTGCACCGGATGT
> Toll 6 F: ATCGGCCAAAAAGAGCAGTA
> Toll 6 R: AGCAGCGTGTGCAGATTATT
> Toll 8 F: ATGCGTACCATTTTCCTACCA
> Toll 8 R: TTTTTACTCTTCGCTTTTTCGC
> Snα-F: CGGAACCGAAACGTGACTAT
> Snα-R: GCGGTAGTTTTTGGCATGAT

Primer pairs were used to amplify a PCR product from genomic DNA. PCR products were directly used in a transcription reaction with T7 polymerase using the MEGAscript transcription kit (Ambion). The reaction was placed in boiling water and allowed to cool to room temperature to promote annealing. RNA was extracted with phenol:chloroform, washed with ethanol, and resuspended in 0.1 M TE buffer.

Stage 2 embryos were dechorionated in 50% bleach for 2 min. The embryos were mounted onto glass slides and desiccated for 4 min using Drierite. Embryos were covered with a 3:1 mixture of halocarbon 700/halocarbon 27 oils and then injected laterally. Embryos were imaged 3 hr after injecting.

## Drug injections

Embryos were dechorionated in 50% bleach for 2 min. The embryos were mounted onto glass slides and desiccated for 4 min using Drierite. Embryos were covered with a 3:1 mixture of halocarbon 700/halocarbon 27 oils and then injected laterally at the beginning of cephalic furrow formation. To disrupt F-actin we resuspended CtyoD (Enzo Life Sciences) at 0.125 mg/mL in DMSO. Embryos were imaged 3–5 min after injecting.

## Laser ablation

Laser ablations were performed using a two-photon Mai-Tai laser set to 800 nm on an LSM710 confocal microscope (Zeiss) through a 40×/1.1 objective, using the Zen software (Zeiss) (WM Keck Microscopy Facility, Cambridge, MA). For ablations, laser power was set at 20%, with a scan speed of 1.58 ms/px. Ablations were performed with four overlapping linear ROIs around one side of MD1.

Sham ablations were performed with the same setup as the laser ablations except the laser power was set at 10%. This caused the ROIs to become bleached, but not ablated.

## Image processing and analysis

All figure images were processed using MATLAB (MathWorks) and Fiji (http://fiji.sc/wiki/index.php/Fiji). A Gaussian smoothing filter (kernel = 0.75 pixel) was applied to all figure images. Brightness and contrast were adjusted to optimally visualize the mitotic spindle or Pins crescents in figure images. Control and experimental images were adjusted the same.

## Division angle analysis

Division angle was taken using the angle tool in Fiji and measured as the angle between spindle poles axis and the AP axis in mitotic cells at anaphase or later. The angle values range from 0°, indicating alignment along AP, to 90°, indicating alignment along DV. Unless otherwise stated, 10 embryos and between 100 and 250 spindles were counted of each genotype. Resulting angles were plotted in rose plots, with percentage of spindles on the r axis and angle on the theta axis.

## Intensity profile analysis

Intensity profiles were taken using the segmented line tool in Fiji with a line width of 2 and measured using the Plot Profile feature in Fiji. The perimeter of the cell was outlined starting at the apical most point of mitotic cells in anaphase. The direction of the lines was toward the dorsal midline or ventral midline (clockwise for the left half of the domains and counterclockwise for the right half of the domains). Intensity was normalized to cytoplasmic intensity, which was measured as the mean gray value of a line drawn through the cytoplasm. Unless otherwise stated, 10 cells per embryo and 3 embryos per genotype were analyzed, averaged, and graphed as line plots with normalized intensity on the y axis and distance from start on the x axis.

For statistical analysis, we found the point of maximum Pins intensity and averaged the troughs to either side of this point to get a peak-to-trough ratio for each cell and then compared this to the ratio achieved by using the corresponding points for Gap43. In experiments with control and experimental, the peak-to-trough ratio was found for the control and compared to the ratio achieved by using corresponding points in the experimental condition.

## Anisotropy analysis

Aspect ratios of interphase cells were taken using the polygon tool in Fiji and the Fit Ellipse measurement, which gives the major axis length, the minor axis, and the angle of the major axis. Aspect ratio is ratio between the major axis and the minor axis. The average aspect ratio was recorded and the angle of the major axis was compared to the axis of division. Unless otherwise stated, 10 cells per embryo and 3 embryos per genotype were analyzed.

## Quantification and statistical analysis

Statistical analyses were performed using the MATLAB statistics toolbox. p-Values were calculated using a two-sample Mann-Whitney test, the reference sample is the angle distribution in wild-type. Correlation coefficients r are Spearman correlation coefficient. p-Values are calculated against the null hypothesis r=0.

# Acknowledgements

We thank members of the ACM lab for helpful discussions on this work and comments on the manuscript. We thank Apolonia Gardner for examining PCP protein localization and coding assistance. We also thank the anonymous reviewers for their feedback. This work was funded by NIGMS R01GM105984 and R35GM144115 to ACM. In addition, JC was supported by NIH Pre-doctoral training grant T32GM007287. This material is also based upon work supported by the National Science Foundation Graduate Research Fellowship under Grant No. 2019275094.

# Additional information

## Funding

| Funder | Grant reference number | Author |
| --- | --- | --- |
| National Institute of General Medical Sciences | R35GM144115 | Adam C Martin |
| National Institute of General Medical Sciences | RO1GM105984 | Adam C Martin |

The funders had no role in study design, data collection and interpretation, or the decision to submit the work for publication.

## Author contributions

Jaclyn Camuglia, Conceptualization, Data curation, Formal analysis, Investigation, Methodology, Validation, Visualization, Writing – original draft, Writing – review and editing; Soline Chanet, Conceptualization, Investigation, Methodology; Adam C Martin, Conceptualization, Funding acquisition, Investigation, Methodology, Project administration, Supervision, Writing – review and editing

## Author ORCIDs
Jaclyn Camuglia (iD) http://orcid.org/0000-0001-9268-9471
Soline Chanet (iD) http://orcid.org/0000-0002-4065-9411
Adam C Martin (iD) http://orcid.org/0000-0001-8060-2607

## Decision letter and Author response
Decision letter https://doi.org/10.7554/eLife.78779.sa1
Author response https://doi.org/10.7554/eLife.78779.sa2

## Additional files

### Supplementary files
- MDAR checklist
- Supplementary file 1. Fly stocks and crosses used in this study.

### Data availability
Data generated and/or analyzed during this study was uploaded to Dryad (doi:https://doi.org/10.5061/dryad.f4qrfj6zb).

The following dataset was generated:

| Author(s) | Year | Dataset title | Dataset URL | Database and Identifier |
|---|---|---|---|---|
| Martin AC, Camuglia J, Chanet S | 2022 | Morphogenetic forces planar polarize LGN/Pins in the embryonic head during Drosophila gastrulation | https://dx.doi.org/10.5061/dryad.f4qrfj6zb | Dryad Digital Repository, 10.5061/dryad.f4qrfj6zb |

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
