## [Editor Report]

This study provides compelling in vivo evidence that mechanical tension emanating from morphogenetic forces during gastrulation orients the spindle at distant sites. The finding that gastrulation-induced forces are required for asymmetric localization of an important and evolutionarily conserved spindle orientation factor, Pins, will be of broad interest to cell and developmental biologists.

---

## [Decision Letter]

[Editors' note: this paper was reviewed by Review Commons.]

---

## [Author Response]

Reviewer #1 (Evidence, reproducibility and clarity (Required)):Camuglia, Chanet and Martin investigate the mechanisms that control cell division orientation in vivo, using the mitotic domains (MDs) in the head of the *Drosophila* embryo as their main model system. They find that cells in the head mitotic domains rotate and align their spindles within 30 degrees of the anterior-posterior axis of the embryo. The Pins protein, implicated in spindle orientation in other systems, is planar polarized in mitotic cells. Pins polarization precedes spindle rotation and is correlated with the division angle (but cell shape is not, violating Hertwig's rule). Overexpression of myristoylated Pins results in uniform Pins distribution on the membrane and affects spindle orientation. α-catenin RNAi (but not canoe RNAi) disrupts Pins polarity and spindle orientation in MDs 1, 3 and 5. Low dose CytoD injections (which should disrupt force transmission) also result in defective Pins polarity and spindle orientations. Finally, mechanical isolation by laser ablation also disrupts spindle orientation. The authors find that preventing mesoderm invagination by snail dsRNA disrupts Pins polarity and spindle orientation in the head.Major1. Is there a certain chirality in the rotation of the spindles? From Video 1, it seems like in MDs 1 and 3 at least, a majority of spindles on the right side of the embryo rotate clockwise, while spindles on the left side rotate counter-clockwise? Is that so, and in that case, are there geometric/molecular considerations that could explain that chirality?

We thank the reviewer for pointing this out. They are correct in that there is a tilt to the spindle orientation relative to the AP axis. To illustrate this tilt, we performed our spindle analysis separately on the right and left sides of MD1 and found that spindles on the left side align with an average division angle of about 30° from the AP axis whereas spindles on the right side align with an average division angle of -30° from the AP axis. To determine whether spindles on either side rotated with a certain chirality, we measured the fraction of times spindles rotated clockwise or counterclockwise on the left and right sides of the embryo and found there was no preference (on the left side of MD1 53% of measured spindles rotated counterclockwise and 47% rotated clockwise, on the right side 46% rotated counterclockwise and 54% clockwise). We have added this data as Figure 1I-J and discussed in the Results lines 135-147.

2. The authors are experts in mesoderm invagination, and understandably concentrate on the role that forces from that process may have in the orientation of head MD divisions. However, the cephalic furrow forms much closer to the head MDs, and in an orientation that might also explain the alignment of spindles in the head. Is cephalic furrow formation important for Pins polarity and spindle orientation in the head MDs?

This was certainly a possibility, but our experimental results argued that mesoderm invagination is most relevant. For example, neither α-catenin or Snail depletion block the cephalic furrow, but do block mesoderm invagination. Both α-catenin and Snail depletion strikingly disrupted spindle orientation and Pins localization, which suggests mesoderm is most important. Furthermore, light sheet imaging of the *Drosophila* embryo has shown that the head region of the embryo undergoes tissue movement in the direction of the cell division and that this is associated with mesoderm invagination (Stern et al., 2022).

To further test the importance of mesoderm invagination, we performed additional ablation experiments to disrupt forces transmitted to the mitotic domains from distinct embryo regions. Consistent with our original hypothesis and previous data, we discovered that separation of MD1 from the cephalic furrow region had no effect on spindle orientation, but that separation from the ventral side of the embryo had a dramatic disruption of spindle orientation. We added this data as Figure 6O-P and included Supplemental Video 7 showing the result. We added to our summary of the data and how it led us to our conclusions in the first paragraph of the Discussion and a model figure illustrating the distinct cuts and their effects (Figure 7I).

3. Does expression of myristoylated Pins afect mesoderm invagination (or cephalic furrow formation)? From Table S1 it seems that a maternal Gal4 driver was used to express myristoylated Pins, which could affect other tissues in the embryo. So it is in principle possible that effects of myristoylated Pins on mesoderm internalization/cephalic furrow formation could affect cell division orientation much like sna loss of function does, but in a mechanism that does not depend on Pins polarity. There is definitely an effect on mesoderm invagination in α-catenin RNAi (but not in canoe RNAi) embryos, so I wonder if the effect could be consistently through defects in mesoderm invagination (or cephalic furrow formation), and Pins polarity is really dispensable for spindle orientation. Are there head-specific Gal4 drivers that could be used to drive myristoylated Pins exclusively in the head?

We apologize that we did not clarify this in our initial submission. Maternal overexpression of myrPins does not obviously disrupt mesoderm internalization or cephalic furrow formation. But, we do see that targeted disruption of mesoderm internalization via a Snail depletion affects the orientation of division. Note that our paper demonstrates the effect of force transmission on Pins polarity and division orientation, which is new and the main conclusion.

In response to this comment we: (1) added language in the Results that states that gastrulation proceeds in myr-Pins expressing embryos (lines 208-210), and (2) Added to the Discussion of the role of these oriented divisions to morphogenesis (lines 442-458). The role of division orientation and division in general in *Drosophila* early embryo morphogenesis is more complicated and will require considerably more work to understand that is beyond the scope of this study.

4. Related to the previous point, does mechanical isolation by laser ablation (Figure 6I-N) affect Pins polarity? This experiment could alleviate some of my concerns above, as it certainly does not (should not?) disrupt neither mesoderm invagination nor cephalic furrow formation.

We agree that it would be useful to look at Pins polarity in laser ablated embryos. Currently, we have been unable to analyze Pins polarity after laser ablation, because the ablation to fully isolate the mitotic domain bleaches fluorescent proteins, especially our Pins::FP signal, which is weaker that other expressed fusions. Also, we have shown that Pins polarity is disrupted by (1) α-cateninRNAi, (2) low dose CytoD injection, and (3) Snail depletion, all of which are expected to disrupt force generation and transmission through tissues.

In response to the reviewer comment, we have determined that Pins::FP still cannot be localized in directional laser ablations, but directional ablations that separate ventral from dorsal MDs cause similar spindle misorientation as Snail depletion, which we show has a striking effect on Pins polarity. Again, remember that myr-Pins does not affect mesoderm internalization and that Snail depletion affects Pins polarity, so we have shown a clear Pins requirement independent of effects on morphogenesis.

Minor1. Figure S5: I am a bit confused about the role of Toll 2, 6, 8 in orienting spindle orientation. In Figure S5D it seems that dsRNA treatment against these genes does not disrupt spindle orientation, but Figure S5F shows quite a significant (p=0.0057) effect in triple mutants. The authors favor the idea that Toll receptors do not affect spindle orientation, but the difference with the mutant should be addressed. Furthermore, what happens in MDs 3, 5 and 14 (if the germband extension defect does not affect those divisions)? Is there a difference between dsRNA and triple mutant embryos in these other MDs?

We think this is a great point. We stated in the text that TLRs are not solely responsible (line 251) for spindle orientation as they do not recapitulate the random pattern of division seen in the myr-Pins expression condition. We acknowledge the differences between the dsRNA injection and TLR triple mutant in the manuscript (lines 242-247), but our data show a greater importance for the role of force transmission. We favor the idea that other mechanisms contribute to spindle orientation because even though the TLR triple mutant had a significant effect on spindle orientation, it did not recapitulate the myristoylated Pins phenotype. In addition, we did not observe planar polarized Bazooka or myosin in the dorsal head, indicating the mechanism detailed by the Zallen lab (Pare et al., 2014) is likely not in play within the mitotic domains. Finally, as described above, we now show that isolating MD1 specifically from the ventral domain is sufficient to entirely disrupt spindle orientation (Figure 6O-P).

2. No statistical analysis is provided for any of the differences in polarity between Pins and Gap43, and this should be done to demonstrate the significance of the polarization of Pins. Also, particularly for MD14, they should compare anterior vs. posterior polarity, as based on the images in Figure 2H it is not clear that there is a difference between the anterior and posterior side of cells.

We thank the reviewer for this point. We have added the statistical comparison.

3. Figure 2A-D: the authors propose that Pins localizes preferentially to the posterior end of cells (instead of both anterior and posterior ends) in MDs 1, 3 and 14 (and anterior in MD 5). How is the asymmetry in the distribution of Pins along the AP axis accomplished, and is there any significance to it? This should be discussed in a bit more detail (currently no potential mechanisms provided in the discussion, just an acknowledgment of the question).

We agree the localization of Pins to the posterior end of cells in MDs 1, 3, and 14 and anterior end in MD 5 is of great interest. We added an acknowledgment of the question and discuss possible models that could explain the result (lines 467-469). Further investigation of the mechanism of this preferential localization will require extensive molecular and genetic studies and is beyond the scope of this paper.

Typos1. Line 49: "one daughter cells" should be "one daughter cell".2. Line 193: "rotation. (Figure 3E-F)." should be "rotation (Figure 3E-F)."3. Lines 232-237: please review.4. Line 238: "epithelia cells" should be "epithelial cells".

We thank the reviewers for carefully reading our manuscript. We have fixed these typos.

Reviewer #1 (Significance (Required)):This is the first study to my knowledge that demonstrates the role of mechanical forces in polarizing Pins, and provides a nice model to further investigate how mechanical forces generated in one tissue may affect cell division orientation in distant ones. The paper is clear, well written, and quantitative analysis is present for most results. I have some issues with the statistics (or lack thereof) for a couple of results, and potential alternative interpretations for some experiments that in my opinion should be addressed prior to publication. Specifically, it is not clear to me if Pins polarity is at all necessary for spindle orientation in any of the examined MDs.Reviewer #2 (Evidence, reproducibility and clarity (Required)):Overview: In this manuscript, Camuglia et al. show Pins/LGN, which is understood to drive spindle orientation, can localize asymmetrically (with respect to the tissue plane) in the *Drosophila* embryo. Experimental work (including drug treatments, laser ablation, and knockdowns) lead the authors to propose that this asymmetry is driven by tissue-level tension. The findings are quite interesting and the manuscript is well-written overall.Major Comments:The authors propose that localization is driven by tissue-level tension, but the direction of the tension isn't clear from the experimental work. For example, the laser ablation experiments cut around the entire perimeter of the mitotic domain, rather than along just one tension axis. Similarly, the finding that disruption of the ventral furrow (by Snail RNAi) interferes with spindle orientation in the head is very puzzling; the furrow is (A) outside the embryonic head and (B) runs in the parallel direction to the divisions considered. The authors need to address the directionality of tension experimentally.

We thank the reviewer for this comment and agreed that better defining the direction of tension would strengthen our manuscript. We showed that blocking mesoderm invagination with Snail depletion disrupts spindle orientation, despite Snail not being required for cephalic furrow formation. Light sheet data has shown that mesoderm invagination is associated with global movements throughout the embryo (Stern et al., 2022; Rauzi et al., 2015; Lye et al., 2015). Furthermore, the ventral furrow extends into the head region just past the anterior of MD5. To address the reviewer’s comments, we: (1) Performed directional laser ablations to determine the directionality of the tension that orients the spindle, and (2) Added to our Discussion more about what is said in the literature about the movements that occur in the head during mesoderm invagination (lines 456-459).

As acknowledged in the text, the asymmetric enrichment of Pins in MD14 is fairly weak. Since the cells being examined here border a divot in the tissue, and might therefore be curving relative to the focal plane, it would be good to rule out the possibility that some of the asymmetry in Pins intensity is just a consequence of cell/tissue geometry. One way this could be achieved is by showing multiple focal planes.

Good point. We do not think that the asymmetric Pins enrichment in MD14 is due to tissue geometry or junction tilt. (1) MD14 divides ~10-15 minutes after mesoderm invagination is completed, so the cells do not border a divot (as seen with Gap43::mCh, Figure 2I). The cells do round up, which can be seen as gaps between cells (Figure 3E). (2) We compared Pins to GapCh and only see an enrichment with Pins (Figure 2H-K). If the enrichment was due to tissue curvature or junction orientation relative to imaging axis, we would see the same enrichment in GapCh. (3) Expression of myr-Pins randomizes spindle orientation in MD14 (Figure 3M, N).

In Figure 3I (and 3M?), it appears that there are fewer cell divisions in the presence of myr-Pins. Is this the case? Since cell shapes change during division, and cell shapes influence tissue tension, an increase in cell divisions could lead to a change in tissue tension. This would be important to address, since tissue tension plays an important role in the proposed model.

These images are not taken at the same point of MD1 division ‘wave’, there are the same number of divisions in each condition. These mitotic domains exhibit a ‘wave’ of cell division (Di Talia and Wieschaus, 2012), and so the number of divisions in each image reflect the timing at which we captured the image. Quantifications involved divisions throughout this wave, but we chose images for figures which are most representative of what we observed. We added this to the text in the final version of the manuscript (lines 207-210).

The α-catenin and Canoe results are a bit confusing:– The rose plot in Figure 4D doesn't show a random distribution of spindle angles, but rather a modest change; most spindles still orient in the normal range. The p value in the figure legend (0.0012) is very different from the one in the figure (5.8284e-04).– Α-catenin is the strongest way to disrupt AJs, but (A) the epithelium appears to be intact in the knockdown condition and (B) spindle orientation is impacted but not randomized. Does this mean that the knockdown is incomplete? Or is Cadherin-mediated adhesion (in which α-catenin participates) only partially responsible for force transduction?

We acknowledge that perturbation using αcat RNAi does not recapitulate the complete disruption of division orientation seen in embryos expressing myr-Pins. This is likely due to the variability in the strength of RNAi knockdown, which is observed for most of the RNAi lines that we use. To address the reviewer’s comment, we added rose plots for individual embryos showing extremes in the severity of division orientation disruption (Figure 4E and F). For the main plot (Figure 4D), we have included all the data because we obviously did not want to pick and choose which embryos were used for analysis. So Figure 4D includes all the variability.

Given that previous studies implicate Canoe in Pins localization, it seems important to lock down the question of whether Canoe is participating in the mechanism described in this paper. How do the authors know the extent of Canoe knockdown? As suggested by the α-catenin results (described above), is it possible that Canoe knockdown is simply not strong enough to impact spindle orientation?Aren't there genetic nulls available?

We thank the reviewer for bringing these points to our attention. There are certainly genetic nulls available (Sawyer et al., 2009), but the experiment suggested by the reviewer would not establish the necessity of Canoe in mitotic domain cells. This is because Canoe nulls severely disrupt mesoderm invagination more severely than RNAi (Sawyer et al., 2009; Jodoin et al., 2015), as well as affecting junctions in the ectoderm during germband extension (Sawyer et al., 2011). Therefore, we would not be able to distinguish what effect of Canoe would be responsible for the spindle orientation using a null mutation. We did better experiments, we used (1) a gene depletion that specifically compromised mesoderm invagination (*snail),* (2) laser ablation to show the importance of force transmission between mesoderm and mitotic domains, and (3) RNAi to deplete Canoe so that mesoderm invagination initiates and pulls on the ectoderm, but where there is clearly compromised Canoe function. This treatment did not cause any effect on spindle orientation arguing against a role of Canoe in this case. In response to the reviewers comment, we added language to the Results to indicate that it is possible that the Canoe knockdown is not strong enough and our rationale for why we did not perform the experiment in a Canoe null (lines 278-280).

Minor Comments:It can be difficult to interpret some of the spindle orientation data since the AP axis is vertical in the diagrams but horizontal in the rose plots. Can one of these be flipped so they go together?

We thank the reviewer for this suggestion and have flipped the rose plots so they match the images. Note that because of the large size of the figures, we have had to consistently orient anterior towards the top, which we establish at the beginning of the Results.

Figure S3 is important information for the reader and should be ideally moved into the main paper. – Protein localizations referred to in text should be annotated on images, as they can be hard to see.

We respectfully disagree that S3 should be included in the main paper. The myr-Pins reagent has been used previously so the information in S3 is not new (Chanet et al., 2017).

There are some discrepancies between figures, legends and text.– p-values differ between figures, legends, and/or text.– Fluorescent markers are labelled differently in figures and legend (CLIP170 in Figure 1)– Graphs appear to show that MD3 polarizes on posterior side, but figure legend says anterior in Figure S1. Vice versa for MD5.

We thank the reviewer for catching these typos. We have fixed these issues.

Ideally, multichannel image overlays should be shown along with individual channels (b/w). However, it is appreciated that the fluorescent signals are exceptionally weak in this study, presenting a challenge to presentation and to quantification.

We agree the overlays would be nice. However, (1) the Pins::GFP signal is weak compared to the tubulin and Gap43 signals, (2) the merge does not provide more clarity, and (3) the figures are already quite large. Therefore, we have only included the separated the images.

Graph axes depicting spindle orientation would be more clear if shown in degrees, instead of normalized or in radians.

We thank the reviewer for this suggestion. We have changed the graph axes to be in degrees.

Reviewer #2 (Significance (Required)):Several recent studies have demonstrated that division orientation (in the tissue plane) is governed by tissue level tension. Remarkably, it appears that diverse mechanisms link tension with spindle orientation. Here the authors provide the first in vivo evidence connecting tension to the asymmetric localization of Pins, an important and evolutionarily conserved spindle orientation factor.Reviewer #3 (Evidence, reproducibility and clarity (Required)):This beautiful manuscript uncovers a role for planar polarized PINS/LGN in orienting the mitotic spindle in *Drosophila* epithelia. In response to morphogenetic forces acting on adherens junctions, PINS/LGN localises to junctions in a planar polarized fashion to orient the spindle, and de-polarization of PINS/LGN prevents planar spindle orientation. The experiments are very well performed and the findings are robust. The conclusions are well supported by the data.Reviewer #3 (Significance (Required)):These important findings mirror previous work in human cell culture, but crucially reveal that the same phenomenon occurs in vivo in the *Drosophila* embryo. Thus, the findings underscore the highly conserved nature and in vivo relevance of this phenomenon.

We thank this reviewer for reading the manuscript and their encouraging words.